

# Spatial assessment of erosive processes in a badland catchment using diachronic LiDAR, Draix, Alpes de Haute-Provence, France

Yassine Boukhari[1], Antoine Lucas[1], Caroline Le Bouteiller[2], Sébastien Klotz[2], Gabrielle Chabaud[1], and Stéphane Jacquemoud[1]

[1]Institut de physique du globe de Paris, Université Paris Cité, CNRS, 75005 Paris, France
[2]Univ. Grenoble Alpes, INRAE, CNRS, IRD, Grenoble INP, IGE, 38000 Grenoble, France

**Correspondence:** Yassine Boukhari (boukhari@ipgp.fr) and Antoine Lucas (lucas@ipgp.fr)

**Abstract.** With denudation rates locally exceeding one centimetre of fresh marl per year, i.e., more than 250 $T.ha^{-1}.yr^{-1}$, the badlands of the Durance basin in the French Alps makes it one of the world's most heavily eroding areas. Since 1983, the Draix-Bléone Observatory has been using hydro-sedimentary stations to instrument several of these small, unmanaged badland catchments, where the hydrological response to seasonal storms is rapid and intense. We combine such chronicles

at the outlet of the Laval basin (86 ha) with a six-year diachronic analysis of airborne and UAV LiDAR data and a bulk density modelling to map mass movements and constrain a catchment-scale mass balance. We find out that landslides and crests failures represents very active areas, accounting for at least 15% of the sediment budget of the watershed, while affecting only 1% of the bare surfaces. They contribute to making the low drainage areas the places with highest erosion rates, reaching as much as two centimetres of fresh marl per year, 3.5 times more than the average value on denuded slopes. Despite some

methodological constraints, our approach seams very promising at quantifying and localising the erosion hotspots as well as assessing sediment transport through critical zone compartments, and could be adapted to time series for monitoring the dynamics of badland catchments in a changing climate.

## 1  Introduction

Badlands are highly erosive landforms with a dissected, ravined-like morphology, largely devoid of vegetation (Bryan and Yair,

1982; Harvey, 2004). They generally develop in semi-arid zones and, to a lesser extent, in humid and sub-humid regions, where the lithology is fragile and highly sensitive to climatic events (Gallart et al., 2002, 2013). The Draix badlands, in the southern French Alps, is one such area. While some hillslopes have been reforested at the end of the XIXth century (Le Brusquet catchment), others are mainly unvegetated and subject to particularly high erosion rates, such as the Laval catchment, where this study is conducted (Mathys et al., 1996).

Numerous studies have been carried out on these badlands at the plot scale in order to analyse the interactions between rainfall, runoff and erosion under controlled conditions and, in particular, to describe the hydro-sedimentary processes associated with Hortonian runoff or subsurface infiltration (Wijdenes and Ergenzinger, 1998; Mathys et al., 2005; Garel et al., 2012). In parallel, high-resolution topography (HRT) acquisition methods are increasingly available to geomorphologists (Lague et al.,





2013; Neugirg et al., 2015; Passalacqua et al., 2015). Using a terrestrial laser scanner (TLS), Bechet et al. (2015) were able to observe regolith swelling, crack closure, micro landslides and the initiation of miniature debris flows (MDFs) at the millimetre scale on such plots. It should be noted, however, that the analyses of the processes may be biased if their size is insufficient in relation to the average distance travelled by the entrained materials (Kinnell, 2009; Boix-Fayos et al., 2006; Yair et al., 2013). Consequently, the experiment carried out by Bechet et al. (2015) was then reproduced (Bechet et al., 2016) on a small 0.13 ha catchment, the Roubine, adjacent to the Laval, allowing the seasonal dynamics of erosion to be observed in detail, in a transport-limited regime in winter and in a supply-limited regime in summer. Similar studies were conducted using TLS surveys in the Spanish Central Pyrenees (Vericat et al., 2014; Nadal-Romero et al., 2015). Yamakoshi et al. (2009) also used a time-lapse camera to monitor the evolution of MDFs during flash floods in the Roubine gully, analysing their characteristics and confirming their essential role in sediment transport at the onset of a flood. Apart from these notable exceptions, studies conducted at the catchment scale in the Draix area have generally focused on investigating and modelling the complex relationship between sediment export and climatic variables (Mathys et al., 2003; Badoux et al., 2012; Taccone et al., 2018; Carriere et al., 2020; Ariagno et al., 2022; Roque-Bernard et al., 2023) or (re)vegetation (Rey, 2003; Burylo et al., 2011; Erktan et al., 2013; Carriere et al., 2020). However, the scaling of erosion mechanisms is highly non-linear with increasing drainage area, due to competing effects between increased gully connectivity and increased sediment storage, and a change in slope distribution as the dominant control of erosive processes (De Vente and Poesen, 2005; Puigdefábregas, 2005; Vanmaercke et al., 2011). For example, the Laval catchment (86ha) and the Roubine catchment (13ha) are neighbours, sharing similar environmental conditions and forcings, but the Laval sediment flux is dominated by suspended sediment contribution, whereas the Roubine sediment flux is dominated by bedload contribution (Draix-Bleone Observatory, 2015; Klotz et al., 2023). To assess the relationship between drainage area and sediment production, Nadal-Romero et al. (2011) analysed 16,571 annual export values at plot and catchment scale from 87 Mediterranean badland sites. They observed a very high and extremely variable sediment production for drainage areas < 10ha, followed by a power-law decrease with drainage area for larger areas. This deviation from the De Vente and Poesen (2005) model for Mediterranean environments shows that badlands, while seemingly ideal natural laboratories, have a intrinsic complexity (Yair et al., 2013; Nadal-Romero and García-Ruiz, 2018). This result was obtained in configurations with strong vegetation and climatic contrasts and different monitoring methods, some of which, such as gauging stations, are considered more reliable than others that are more widely used, such as runoff plots or erosion pins (Nadal-Romero et al., 2011). This calls for a multi-scale study that integrates, for comparative purposes, several scales within the same catchment, using a unique measurement method.

Airborne remote sensing methods are often used to inventory local sediment sources, assess their volume and discuss their role in the overall erosion of a catchment (Guzzetti et al., 2012; Bechet et al., 2016; Krenz and Kuhn, 2018; Bernard et al., 2021). In the case of landslides (Parker et al., 2011; Li et al., 2014), this is facilitated by empirical area-volume relationships (Guzzetti et al., 2009; Larsen et al., 2010; Massey et al., 2020; Alberti et al., 2022; Yunus et al., 2023), but also by the advent of photogrammetric methods based on aerial photographs (D'Oleire-Oltmanns et al., 2012; Krenz and Kuhn, 2018; Lucas and Gayer, 2022). More recently, airborne LiDAR systems (Bull et al., 2010; Passalacqua et al., 2015; Okyay et al., 2019; Bernard et al., 2021) have made it possible to reconstruct the topography of an entire basin at high resolution, while limiting




obstructions by slopes (Brodu and Lague, 2012; Stöcker et al., 2015). To date, however, such studies estimate the volumes
of sediment sources and sinks without assessing the corresponding mobilised masses (Krenz and Kuhn, 2018; Bernard et al.,
2021), thus failing to take into account the variations in compaction associated with erosive processes like landslides (Chen
et al., 2005; Bernard et al., 2021). These variations can be significant, especially in badlands where there is a 1:2 ratio between
the bulk density of the colluvial deposits at the foot of slopes and that of the unweathered marl (Mathys et al., 1996).

This study aims to fully explore the potential of diachronic LiDAR data to study erosion processes in the Laval experimental
basin, where both sediment density and production are measured at a gauging station at its outlet. Section 2 describes the study
area and presents the gauging station and LiDAR data. Section 3 describes the procedure for mapping mass movements and
performing a mass balance at the catchment scale. The results examined in Sect. 4 highlight the contributions of the inven-
toried sediment sources and sinks to the erosion dynamics of the watershed. The introduction of specific drainage area, i.e.,
the drainage area per unit flow width (Bernard et al., 2022), allows us to locate these sources and sinks within the hydro-
graphic network and to calculate sediment production rates at different scales in different critical zone compartments. Finally,
Sect. 5 discusses the suitability of our methodology for assessing badland erosion processes at different scales and presents
perspectives for event and seasonal erosion monitoring.

## 2 Study site and data

### 2.1 The Draix-Laval experimental basin

#### 2.1.1 Draix-Laval's *Terres Noires*

The Laval catchment is a marly, torrential watershed of $86\mathrm{ha}$, of which more than 60% are gullies and bare steep slopes
(typically $40° - 50°$) characteristic of badlands. It lies between $850\mathrm{m}$ and $1250\mathrm{m}$ in the Bléone valley, near the town of Draix
and upstream from Digne-les-Bains (Alpes de Hautes-Provence) (Fig. 1).

Draix has a Mediterranean mountain climate, with annual precipitation of about $900 \ \mathrm{mm}$ and considerable interannual
variability ($\pm \ 200 \ \mathrm{mm}$). Harsh winters are conducive to the weathering by frost-cracking processes of the Jurassic black
marls called 'Terres Noires' (Ariagno et al., 2022). Rainfall in spring and autumn is recurrent but not very intense. Storms are
frequent in late spring and summer. Their paroxysmal intensity over short periods (Mathys et al., 2005; Ariagno et al., 2022)
is responsible for torrential floods with concentrations up to $800 \ \mathrm{g/L}$ and event-scale sediment export up to several hundreds
cubic meters (Klotz et al., 2023). This results in a very high inter-annual variability in the sediment export, reaching about
half of the total. On a regional scale, the Terres Noires are responsible for almost 40% of the sediment load of the Durance,
although they represent only 1.2% of its catchment area (Copard et al., 2018).

#### 2.1.2 Draix-Bléone Critical Zone Observatory

The extreme fragility of the Laval black marls and the surrounding basins make them ideal experimental sites for studying
the processes of badlands erosion. This is why the INRAE (Institut national de recherche en sciences et technologies pour



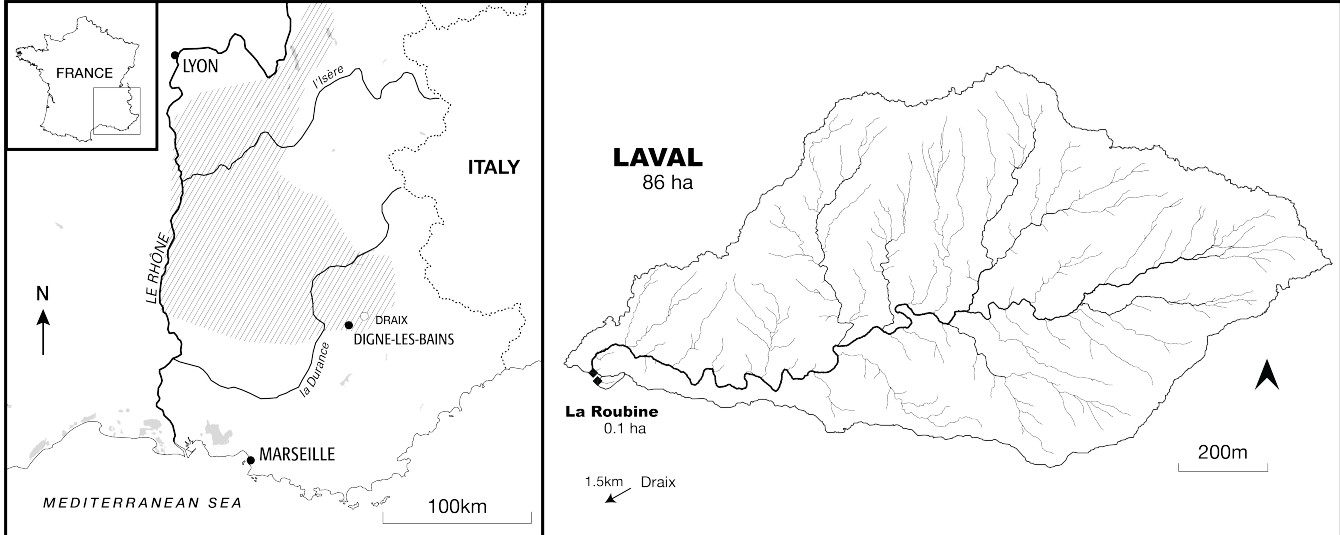

**Figure 1.** Map showing the extent of the "Terres Noires" (adapted from Antoine et al. (1995)) and the Laval basin.

l'environnement et l'agriculture) has been monitoring these basins since 1983, and since 2000 as part of the Draix-Bléone Critical Zone Observatory (CZO). The latter joined in 2015 the OZCAR research network (Gaillardet et al. (2018), www. ozcar-ri.org), dedicated to the study of the critical zone.

A hydro-sedimentary station was installed upstream of the confluence of the Laval ravine and the Bouinenc, a tributary of the Bléone river. It consists of a rain gauge, a water level sensor for indirect measurement of the flow in the Parshall flume, automatic water samplers and a turbidimeter to measure the suspended sediment discharge. The coarsest materials that make up the bedload are deposited in the 1400 $\mathrm{m}^3$ sediment trap, which is emptied once or twice a year. Topographic surveys of this trap are made after each intense event, allowing us to measure the bedload contribution to total sediment export. All data from the Laval station, as well as from the other Draix-Bléone catchments, are described in Klotz et al. (2023) and are available in the BDOH database repository (Draix-Bleone Observatory, 2015).

## 2.2 LiDAR campaigns

Two LiDAR surveys were used to carry out a high spatial resolution diachronic study of the topography of the Laval basin between 2015 and 2021. Table 1 summarises the characteristics of the two point clouds.

### 2.2.1 UAV LiDAR survey (7 April 2015)

An initial survey was carried out on 7 April 2015 as part of a project supported by the OSUG@2020 LabeX. The Laval basin was scanned by the Sintégra company using a RIEGL LMS-Q680i full-waveform LiDAR mounted on a UAV helicopter.



**Table 1.** Point cloud characteristics for the 2015 and 2021 airborne LiDAR campaigns for the whole catchment points and for the ground point subset.

| LiDAR Surveys | Number of Points | | Point Cloud Density | | Typical 3D Distance | |
|---|---|---|---|---|---|---|
| | Catchment | Ground | Catchment | Ground | Catchment | Ground |
| 07/04/2015 | 111,254,755 | 42,432,634 | 98 m$^{-2}$ | 44 m$^{-2}$ | 6 cm | 5 cm |
| 23/06/2021 | 28,300,873 | 15,813,801 | 30 m$^{-2}$ | 20 m$^{-2}$ | 8 cm | 13 cm |

The resulting point cloud is certified, georeferenced, and classified between ground and above ground. The altimetric accuracy is estimated to be 3 cm from GPS measurements taken on a control surface. The nominal planimetric accuracy is said to be 20 cm, but the error is probably less (the nominal altimetric accuracy is 10 cm).

### 2.2.2  Airborne IGN LiDAR HD survey (23 June 2021)

The LiDAR HD programme, led by the IGN (Institut national de l'information géographique et forestière), aims to provide free access to 3D mapping of France (metropolitan France and overseas departments and territories, excluding French Guiana) with an accuracy of 10 cm by the end of 2025 (IGN, 2024). The entire country is not yet covered by the programme, but the Draix Bléone CZO catchment area was observed on June 23, 2021 at 7:34 UTC. The LIDAR is mounted on an aircraft and uses a mirror tilting system to acquire data in bands.

The data is georeferenced in the Lambert 93 coordinate system and cloud segmentation is applied to distinguish ground, vegetation, buildings, etc. using IGN's myria3D deep learning algorithm (Gaydon, 2022). The programme specifications indicate a minimum accuracy (RMS) of 10 cm for altimetry and 50 cm for planimetry.

## 3  Method

### 3.1  Outlet cumulative sediment export

The hydro-sedimentary chronicles are available on the website of the observatory (Draix-Bleone Observatory, 2015). From the instantaneous discharge and suspended sediment concentration, and the volumes scoured from the sediment trap, converted to tonnes at the measured density of 1700 kg.m$^{-3}$ (Klotz et al., 2023), we can estimate the total export of sediment $M_{tot.}$ at the station. Table 2 summarises the annual and cumulative export between the two LiDAR campaigns, with expanded uncertainties (see Appendix C for details). It totals $89.0 \pm 28.6$kT between April 2015 and June 2021.

Taking two similar periods, between April 2008 and June 2014 and between April 2012 and June 2018, we obtain in the same way a total sediment export of $87.0 \pm 26.7$kT and $87.5 \pm 28.8$kT. In the following we consider the former value to be representative of the behaviour of the basin on this timescale, which is long enough to compensate for the large inter-annual variations.



**Table 2.** Annual and cumulative sediment export measured at the catchment outlet between the two LiDAR campaigns. $M_{susp.}$ and $M_{dep.}$ are the cumulative suspended matter and deposited sediment in the sediment trap. $M_{tot}$ is the sum of these contributions. 2015 and 2021 are marked with a † in the table to indicate that data are reported from 7/04/2015 up to 23/06/2021.

| Sediment export (kT) | 2015† | 2016 | 2017 | 2018 | 2019 | 2020 | 2021† | Cumulative |
|---|---|---|---|---|---|---|---|---|
| $M_{susp.}$ | $12.3 \pm 5.4$ | $7.3 \pm 2.7$ | $4.3 \pm 1.1$ | $13.7 \pm 4.5$ | $19.4 \pm 8.8$ | $5.3 \pm 2.6$ | $2.0 \pm 1.0$ | $64.3 \pm 26.1$ |
| $M_{dep.}$ | $4.1 \pm 0.4$ | $3.8 \pm 0.4$ | $2.4 \pm 0.2$ | $5.5 \pm 0.6$ | $4.9 \pm 0.5$ | $2.8 \pm 0.3$ | $1.2 \pm 0.1$ | $24.7 \pm 2.5$ |
| $M_{tot.}$ | $16.4 \pm 5.8$ | $11.1 \pm 3.1$ | $6.7 \pm 1.3$ | $19.2 \pm 5.1$ | $24.3 \pm 9.3$ | $8.1 \pm 2.8$ | $3.2 \pm 1.1$ | $89.0 \pm 28.6$ |

## 3.2 Refinement of the co-registration of campaigns

Co-registration of LiDAR campaigns is a major source of systematic error, as a shift of one centimetre in $z$ between point clouds of an $86$ha catchment will led to an over- or underestimation of the total volume of $8600 \text{ m}^3$. Systematic co-registration errors can also occur in the horizontal plane, leading to ridge misalignment. In this study, we refine the point cloud co-registration by analysing the distribution of local distances on subsets of the catchment area: on nearly stable flat surfaces for vertical errors, and on slopes with simple geometry for ridge alignment (see Fig. E1). We show that a relative shift of about $(\Delta X, \Delta Y, \Delta Z) = (10, 11, 0.5)$ expressed in centimetres, of the order of the distance between neighbouring points within a cloud, must be applied to the 2015 campaign to perform accurate diachronic analysis with the 2021 campaign. The absolute planimetric and altimetric uncertainties presented in Sect. 2.2 are reduced to $(\delta X, \delta Y, \delta Z) = (\pm 5, \pm 5, \pm 1)$ in the relative position between clouds.

## 3.3 Diachronic analysis of local volume change

The evolution of the topography from one campaign to the next is assessed by calculating the local distances between the corresponding clouds along the normals to the source surfaces. This is done using the M3C2 (Multiscale Model to Model Cloud Comparison) method developed by Lague et al. (2013), which takes into account the local roughness scales of complex natural surfaces. By studying LiDAR data and aerial photographs of the Super-Sauze landslide, also composed of Jurassic black marl, Stumpf et al. (2015) have shown that this method is an accurate and versatile tool for analysing these active areas, outperforming point-to-point or point-to-mesh measurements. As the point clouds for the 2015 and 2021 campaigns distinguish vegetation or structures from ground points, only the latter sub-cloud is used in our study for each campaign. Given the complexity of our surfaces and the point densities presented in Table 1, we empirically set the local scale suitable for distance calculation to $r = 30$cm for both clouds.

The mapping of volumetric variations in topography is achieved by rasterising the resulting point cloud to a $1 \text{ m}^2$ grid. This allows small mass movements to be captured while taking into account the density of the point clouds and avoiding empty cells. This gives an average value to the different fields, including height and local distance. From this grid, we construct prisms whose volume corresponds to the local variation between the two campaigns: the surface model is used to orientate the base facets according to the topographic gradient, and the local distance between the point clouds determines their height (Appendix A). The signs of the M3C2 local distances are retained in the volume calculations to indicate whether accumulation or erosion





has occurred. Standard deviations can also be propagated throughout, allowing us to estimate the volume uncertainties of each
irregular voxel.

### 3.4   Effective marl density modelling

Land movements in the catchment can lead to local and transient accumulations of matter, whereas the integrating nature of
the hydrographic network induce that all this matter is ultimately measured at the hydro-sedimentary station. It follows that a
mass balance carried out at the scale of the open system constituted by the catchment should be closed by the export values
measured at the station.

    In order to establish a watershed mass balance and capture the contributions of different erosion modes, it is essential to
map (and sum) changes in mass rather than volume. This is achieved by using local bulk densities, which cannot be measured
directly at the catchment scale and are likely to vary considerably with the local material type (fresh bedrock, regolith, alluvial
or colluvial deposits), which depends on the local history of scour and deposition. A simplified bulk density model, the con-
struction of which is detailed in Appendix B, has therefore been developed based on sediment deposition measurements and
marl weathering profiles from other studies (e.g. Maquaire et al., 2002; Ariagno et al., 2023). Figure 2 shows the effective dry
density as a function of the measured local distance between the 2015 and 2021 topographies. The resulting profile is framed
by two variations by $\pm 0.3$ density points at a given depth, which makes it possible to estimate the extended uncertainties. This
model does not take into account the spatial variability of marl deposition and weathering profiles.

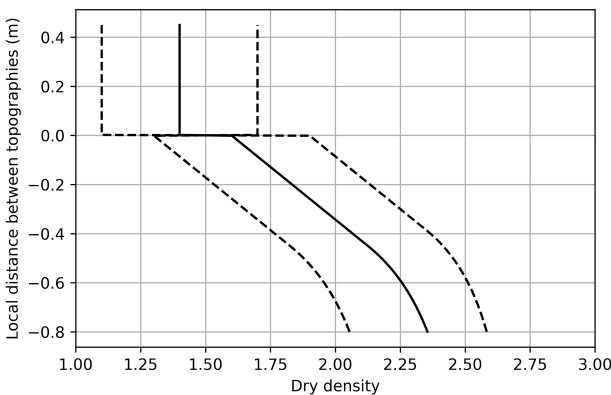

**Figure 2.** Dry bulk density profile $\rho_{eff}(d_\perp)$ for weathered marls and sediment deposits as a function of local distance measured between
topographies.





## 4 Results

### 4.1 Mapping of erosion and deposition signals

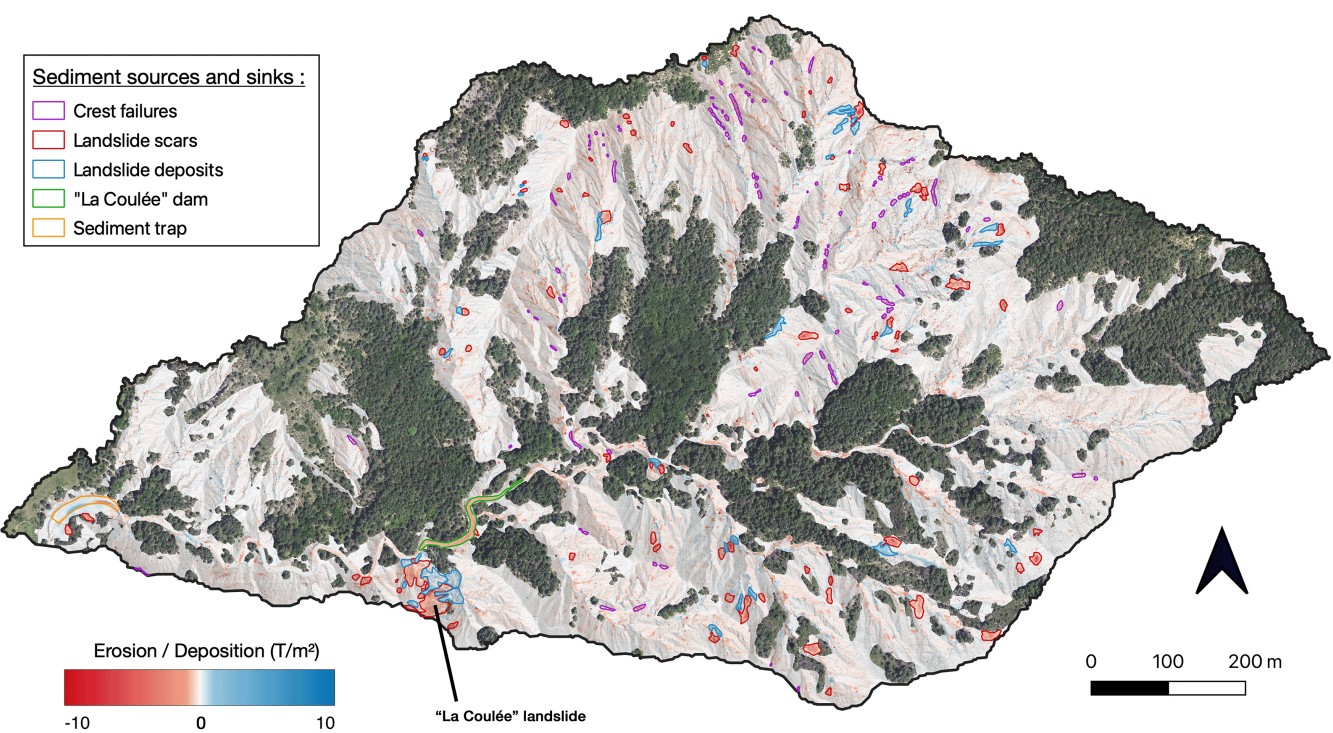

**Figure 3.** Local erosion (in red) and deposition (in blue) mapping between the 2015 and 2021 LiDAR surveys over the Laval catchment (French Alps). Vegetated areas are masked by an orthophotography (Institut Géographique National, 2021). An inventory of the sediment sources and sinks is also overlaid.

Figure 3 shows the local mass changes in the denuded areas of the Laval basin. We exclude here vegetated areas, where erosion is assumed to be negligible compared to denuded areas (Carriere, 2019; Bunel et al., 2025) and where surface reconstruction is considered less reliable (see Sect. 5.1). It appears that the mass variations are not evenly distributed across the basin: 97% of the bare areas have values between $-1\mathrm{T/m^2}$ and $+1\mathrm{T/m^2}$ (about $\pm40\mathrm{cm}$ of fresh marl), but this represents only 54% of the total mass balance on denuded areas. This is because significant signals greater than a few $\mathrm{T/m^2}$ are found on slopes, in areas limited to a few tens of square metres. Erosion and deposition signals are generally associated, with the latter



extending a few metres downstream of the former. These local movements are sometimes visible on orthoimages constructed from aerial photographs, supporting the interpretation of landslides or debris flows. In addition, some of the erosion signals

tend to be located on the ridges of the slopes and are therefore referred to as crest failures. Finally, some strong signals are also found in the main drain, in the sediment trap at the outlet, and 650 m upstream, extending over 200 m. These erosion and deposition hotspots are manually labelled in Fig. 3.

### 4.2  Contribution of erosion modes to sediment production

Calculating the total mass balance at the catchment scale, excluding vegetated areas, we obtain an export estimate of 60±20

kT. With the previous assumptions, in particular the choice of a bulk density profile, we are able to explain about 67% of the sediment export measured at the outlet, which is of the order of 89±30 kT. Uncertainties are discussed in Sect. 4.1.

Sediments produced by landslides account for a significant proportion (14%) of the measured export, but occupy less than 1% (0.8 ha) of the surface area of the basin. However, this proportion falls to 10% if we assume that only 71% of the deposits associated with these landslides are drained on average. Similarly, the erosion observed on ridges accounts for 4% of the export,

covering only 0.2 ha (4 times less than landslides). These estimates could be underestimated as there is a 23% discrepancy in the mass balance, which falls within the uncertainty ranges.

Figure 3 shows that significant erosive activity is associated with the 'La Coulée' landslide that occurred in December 1998 on the left bank of the main channel. This structural landslide mobilised between 4500 and 5600 $m^3$ of compact marl (Fressard and Maquaire, 2009), corresponding to 12 to 15 kT, and temporarily obstructed the Laval torrent. According to Malet (2003)

and Mathys (2006), this landslide, by striking the opposite slope, forced the gradual evacuation of its materials by the torrent. This dynamic continues as the landslide seems to have generated 2.2 kT of sediment between our two surveys, of which 0.7 kT did not reach the main channel and remained on its slopes.

The timescale at which we perform the diachronic LiDAR analysis is such that Fig. 3 integrates several of the seasonal variations observed in the main channels notably by Bechet et al. (2016); Jantzi et al. (2017); Liébault et al. (2022), making

the interpretation more difficult. Nonetheless, the strong signal observed in the channel immediately upstream of the landslide suggests that the obstruction continues to disrupt sediment transport in the basin. It seems that this obstacle has created a temporary reservoir that can be filled under a transport-limited regime or, conversely, emptied under a supply-limited regime.

### 4.3  Localisation of mass variations within the hydrographic network

To further investigate the contributions of each class of sediment sources and sinks in relation to their location within the basin,

we reconstructed the hydrographic network of the Laval catchment under flood conditions using the 50 cm DEM derived from the 2015 LiDAR campaign and the GraphFlood algorithm (Gailleton et al., 2024). The latter efficiently uses graph theory to solve the 2D shallow water equations and thus models the characteristics of the flow (flow rate, water height, flow width) under steady-state conditions for given runoff rates. Here we choose a high rate of about 50 mm/h, corresponding to an intense rainfall that is likely to generate sediment transport. The method allows the introduction of hydro-geomorphic metrics such

as the specific drainage area (Fig. 4), sometimes called the effective drainage area, generally constructed as the ratio of the



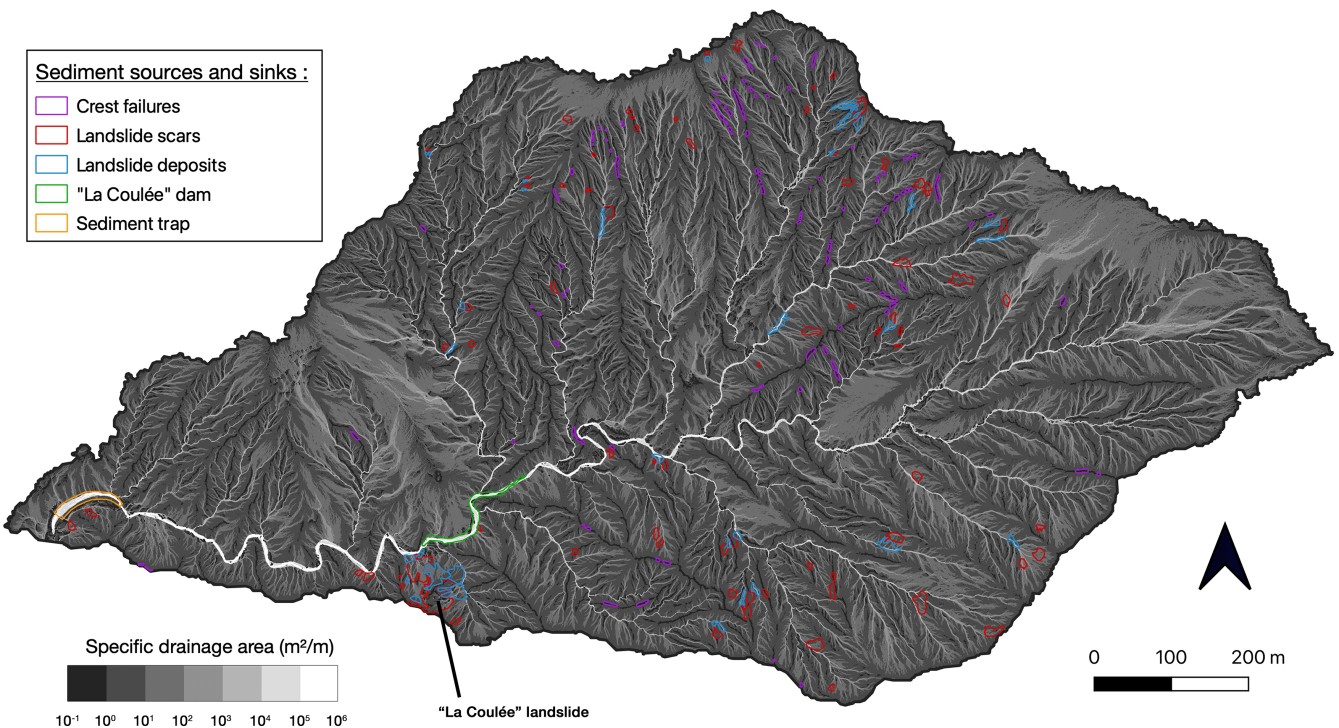

**Figure 4.** Map of the specific drainage areas binned in decades. An inventory of the sediment sources and sinks is overlaid.

drainage area and the flow width estimated from contours and, within this algorithm, as the ratio of the discharge per unit flow width (specific discharge) and the runoff rate (Bernard et al., 2022; Gailleton et al., 2024). It is commonly used for hydrological (Beven and Kirkby, 1979; Bernard et al., 2022) and slope erosion modelling (Moore and Wilson, 1992; Dietrich et al., 1993) or in combination with slope to define the Topographic Wetness Index (TWI) as a proxy for soil moisture (Beven and Kirkby, 1979; Riihimäki et al., 2021). The widespread use of high resolution digital elevation models, in particular those generated

using airborne LiDAR, has made it possible to accurately describe the structure of watercourses and to use this metric, which takes into account flow width (Bernard et al., 2022). Figure 5.a) shows the distribution of local mass variations due to erosion or deposition as a function of the corresponding specific drainage area, indicating the sources and sinks previously identified in Fig. 3. A second scale indicates the correspondence between the specific drainage area and the upslope contributing area (see

Fig. D1).




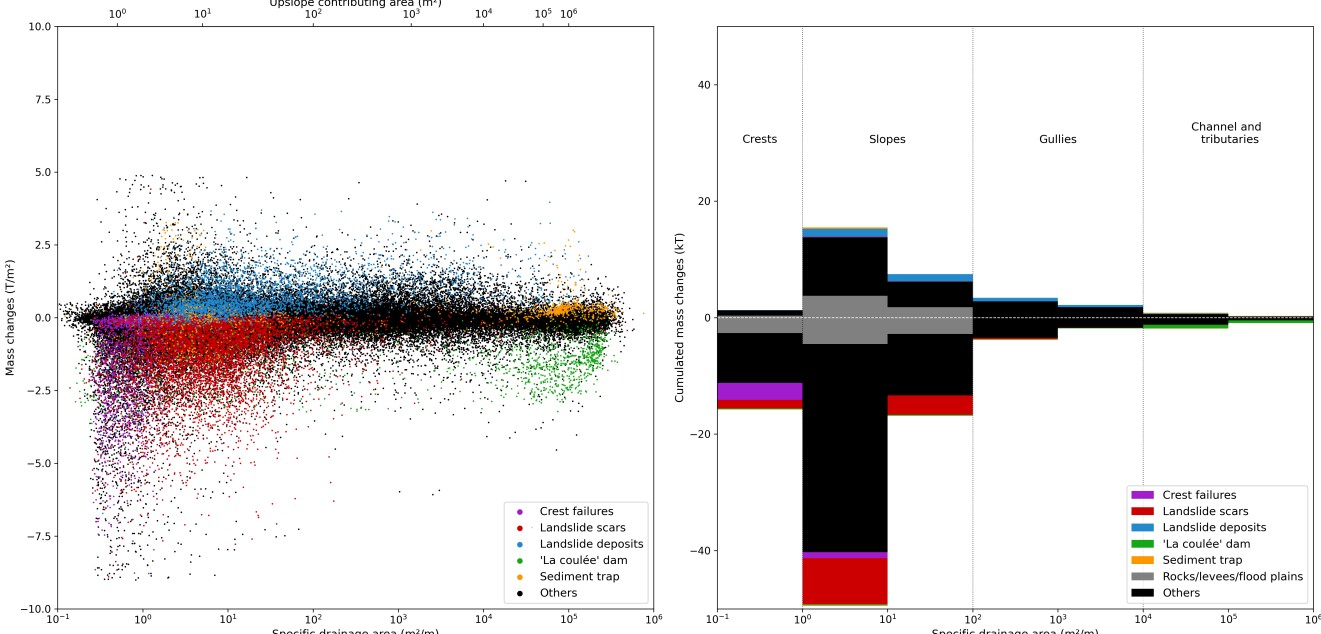

**Figure 5.** Distribution of local mass variations as a function of the associated specific drainage area (left plot) at any point in the denuded regions, (right plot) cumulated in each decade of the specific drainage area. The sediment sources and sinks identified above are shown in the same colours as above. Contributions from suspected levees, emerging rocks and floodplains with small drainage area but belonging to the hydrographic network area are also shown.

As already mentioned, a large majority of points correspond to mass changes between $-1\mathrm{T/m}^2$ and $+1\mathrm{T/m}^2$ and are distributed over all drainage areas. However, almost 50% of the mass balance comes from outside this range. Points with higher deposition are mainly concentrated in the specific drainage areas between $10^0$ and $10^1$ $\mathrm{m}^2/\mathrm{m}$ and to a lesser extent, between $10^1$ and $10^4$ $\mathrm{m}^2/\mathrm{m}$, mostly corresponding to landslide deposits. Points of higher erosion are mainly concentrated in
specific drainage areas below $10^2$ and above $10^4$ $\mathrm{m}^2/\mathrm{m}$, corresponding respectively to crest and landslide erosion, and main channel drainage.

By examining the sums of positive (deposition) and negative (erosion) cumulative mass changes for each decade of specific drainage area, it is possible to determine their weight in the sediment balance, as well as those of the labelled sediment sources and sinks within them (Fig. 5.b). As some low drainage area signals actually correspond to levees, emerging rocks, floodplains
and terraces of the main channel and tributaries, we define a 1m-buffer around gullies and channel above $10^3$ $\mathrm{m}^2/\mathrm{m}$ and a 2m-buffer above $10^4$ $\mathrm{m}^2/\mathrm{m}$, which creates another class in Fig. 5.b). By excluding this contribution, Fig. 4 and 5.b) show that specific drainage areas feed into each other from ridges to slopes and finally hydrographic network so that we now assume that the decades of specific drainage area may actually reflect critical zone compartments that are susceptible to produce and transport sediments with different dynamics:




- The crests typically have submetric specific drainage areas, i.e., upslope contributing areas ranging from 0 to about 1/3 $m^2$. For this decade, 17% of the mass balance corresponds to rocks and levees in the channels, 19% to the previously identified crest failures and about 9% to landslide scars, the last two together occupying 6% of the surface area of the compartment. As expected, there is almost no accumulation in this compartment and the mass balance totals 12 kT, i.e., 20% of the overall balance for less than 7% of the denuded area.

- The "Slopes" compartment are defined by the following two decades of specific drainage area and therefore range up to 100 $m^2$. Over these decades, crest failures are negligible but landslide scars account for 16% and 20% respectively of the mass balances, while deposits account for 9% and 17% (2-3% of the corresponding surface areas). For deposition and erosion, most of the signals are not classified, and probably correspond to diffuse processes such as erosion by Hortonian runoff, creep or erosion/filling or small rills on slopes. Here again, up to 17% of the cumulative mass changes over these decades correspond to levees and flood plains belonging to the hydrographic network. Excluding these contributions, the slope compartment accounts for 69% of the overall mass balance, while occupying 79% of the denuded areas.

- The remaining four decades describe the hydrographic network itself, with gullies up to $10^4$ $m^2$ (1 ha) and main channel and tributaries above that, consistent with the considerations made by Bechet et al. (2016) and Nadal-Romero et al. (2011). Compared to the previous compartment, the signal corresponding to the hydrographic network is small, even in the main channel where we identified a dam upstream of the "La Coulée" landslide (1.3 kT), or taking into account the levees, floodplains and terraces (−4 kT). We could have observed a signal up to 2.4 kT coming from the accumulation beach, but this is not the case as it appears at the same filing level between the two campaigns (Fig. 3).

## 4.4 Production rate calculation for each specific drainage area compartment

Mass balances can be carried out for each compartment or decade of specific drainage areas, taking into account that they feed into each other and may be subject to transient sediment deposition and drainage, as sediment tend to move in pulses accross the landscape (Puigdefabregas et al., 1999). We should also take into account contributions that have low specific drainage areas, but are actually part of the hydrographic network, as explained above.

In order to derive the corresponding production rates, it is necessary to exclude the contributions due to remobilisation. To achieve this, we consider two boundary cases for each compartment, corresponding to the transition from a transport-limited regime to a supply-limited regime, or vice versa. Indeed, the first campaign may occur at a time when erosion is transport-limited, such that sediment, at most the total upstream production, accumulates in a given compartment. These deposits may be drained between the two campaigns, with the second campaign occurring when erosion is supply-limited. The mass balance in this compartment may therefore overestimate the amount of sediment produced inside by as much as the total upstream production, which gives us an initial constraint on maximum sediment production. Conversely, the second campaign may have been carried out at a time when the given compartment is accumulating material produced upstream, under transport-limited conditions, whereas the first campaign was in a supply-limited regime. Again, these deposits may represent at most all of what has been produced upstream, so that when a diachronic analysis is used to derive erosion rates, the production of the





compartment may be underestimated by as much. We are probably closer to the former case, which is consistent with the draining of the so-called 'La Coulée' natural dam (Fig. 3) by late spring rainfalls (31/04, 01/05 and 10/05), as the first survey
took place in early April 2015 and the second in late June 2021.

For crests not fed from upstream, the production rate is simply the ratio of the mass balance for a compartment to the corresponding area. We obtain a value of $-42$ kg/m$^2$/year, i.e., a production rate twice as high as the $-21$ kg/m$^2$/year obtained from the export values measured at the outlet, integrating the production on all of the denuded slopes of the catchment. For the slopes, the previous approach gives a production rate of between $-16$ and $-8$ kg/m$^2$/year, with an indicative value
of $-12$ kg/m$^2$/year, which is already lower than the average value obtained from outlet export, but in line with the $-13$ kg/m$^2$/year obtained with the export measured by our diachronic analysis. Again, these values could be underestimated as their remain a 23% discrepancy in the mass balance. Although our framework could theoretically be applied to gullies and channel tributaries, the resulting uncertainties are too large to provide a reliable estimate of their production rates.

## 5  Discussion

### 5.1  Methodological constraints

To our knowledge, this is the first study to perform a catchment-scale mass balance with sediment export measurements at the outlet (89±30 kT) and mapped mass movements over the bare areas of the catchment (60±20 kT). The performance of this method depends on :

- The quality of the LiDAR time series (point cloud density, accuracy) and its co-registration, assessed over stable zones.
The sensitivity of our global balance to a $z$-shift is estimated to be around 10 kT.cm$^{-1}$, while our uncertainty on $z$ is estimated to be in the millimetre to centimetre range at most (Fig. E1).

- Reconstitution of local variations in volume. The associated uncertainties result essentially from the choice of grid cell size, ±8 kT, depending on whether a size of 0.5 m × 0.5 m or 2 m × 2 m is chosen. The 1 m value limits the number of empty cells for which the value must be interpolated. The other uncertainties propagated in the processing chain are
of the order of a few hundred tonnes at most. In addition, the effect of rainwater infiltration can also cause the regolith to swell or shrink on the millimetre scale (Bechet et al., 2015), which could result in a weak measured signal that does not correspond to erosion.

- The design of a bulk density model to be associated with the calculated volume variations at each point in the basin. This is the most difficult variable to constrain as the spatial and temporal variability of weathering and deposition profiles can
significantly alter our estimates of displaced or accumulated mass (Ariagno et al., 2023; Maquaire et al., 2002; Travelletti et al., 2012). The density ranges defined in Sect. 3.4 give mass balance uncertainty estimates of ±20 kT, i.e., 7 kT per tenth of a density shift.





– Measurement uncertainties at the outlet, both in terms of suspension and deposition in the sediment trap. They mainly reflect the difficulty of calibrating turbidity measurements to estimate suspended matter at low concentrations, as explained in Table C1 (Klotz et al., 2023). They range within $\pm 30$ kT over the period.

Our work has been carried out on bare badland formations, excluding vegetated areas which are likely to have a very different weathering profile compared to the proposed density model. In any case, although they may be the site of deposition or transport, it is expected that material flow will be low due to soil fixation by the root system (Rey, 2003; Burylo et al., 2011, 2012; Carriere et al., 2020; Bunel et al., 2025). In addition, the density of LiDAR points classified as 'ground' is lower under vegetation, making it more difficult to reconstruct the topography on such surfaces that may be littered with plant leaves or schrubs.

Finally, in this study, the inventory of sediment sources and sinks is carried out manually using a GIS tool, with limitations in terms of contour delineation and detection thresholds (Guzzetti et al., 2012). An alternative, although also subject to problems of merging and underdetection (Li et al., 2014; Marc and Hovius, 2015; Tanyaş et al., 2019), could be based on supervised or unsupervised clustering (Borghuis et al., 2007; Parker et al., 2011), similar to the methods used for landslide detection using pixel-based (Mondini et al., 2011; Lu et al., 2019), object-oriented (Stumpf and Kerle, 2011; Keyport et al., 2018) or deep learning (Ghorbanzadeh et al., 2019; Prakash et al., 2020) approaches. Our results suggest that the specific drainage area or the Topographic Wetness Index (Beven and Kirkby, 1979) are good feature candidates, alongside the classical hydro-geomorphological metrics (e.g., elevation, curvature, slope, flow direction).

## 5.2 Spatial and temporal scales

A characteristic of the badlands is the conjunction of sparse vegetation, particularly grasses and shrubs to consolidate the regolith with their root networks, the predominance of steep slopes, exceeding $45°$, and the impermeability of the marls that form the bedrock beneath the weathered layer. This favours the emergence of Hortonian runoff between the gullies, which effectively washes away the altered material produced in winter on the bare slopes (Descroix and Olivry, 2002; Descroix and Mathys, 2003). Therefore, in line with the data analysed by Nadal-Romero et al. (2011), and in contrast to other Mediterranean environments (De Vente and Poesen, 2005; De Vente et al., 2007), our study shows that the areas producing the most sediment in badlands are those with the lowest drainage area and the steepest slopes. In this way, we are able to measure 75% of the export for specific drainage areas smaller than $10 \text{ m}^2/\text{m}$ (about 2/3 of bare areas) and 20% of the export solely for submetric specific drainage areas (less than 7 % of the bare areas). In accordance with the corrigendum (Nadal-Romero et al., 2014) to the study of Nadal-Romero et al. (2011), Fig. 5 shows lower production below $1 \text{ m}^2$ of upslope contributing area (smaller than the sampling scale) where the slope becomes convex, no longer concentrates runoff, and is only characterised by "splash erosion" (impact of raindrops on the weathered regolith). That said, as soon as this limit is exceeded, the efficiency of the erosion processes is such that the crests compartment defined above has a production rate twice as high as that of the slopes.

The timescale of the study also determines the processes to which such a diachronic analysis can be sensitive. The accumulation of signal over several years makes it easier to detect and reliably quantify gradual erosion on slopes that are unlikely to



store sediment temporarily, particularly where the terrain is very steep. It also makes it possible to estimate mean production rates that compensate for inter-annual variability and provide a representative description of the behaviour of the basin on these timescales. Conversely, adapting this method to characterise sediment connectivity in the hydrographic network requires a finer temporal resolution to capture seasonal alternations between a transport-limited erosion regime in winter and a supply-limited 335 regime in summer (Bechet et al., 2016; Ariagno et al., 2022). Together with finer spatial resolution, we should be able to link large mass movements to climatic forcing at the event scale. This could help to better constrain the modelling of sediment transport during floods and to provide explanatory elements for the hysteresis loops, especially clockwise loops observed at hydro-sediment stations (Roque-Bernard et al., 2023), which we suspect may be related to debris flow inputs from slopes or gullies.

## 5.3 Active mass wasting areas

Our study highlights the importance of landslide scars and crest failures as erosional hotspots. As explained in Sect. 4.2, they contribute to about 15% of the sediment budget of the basin, even though they affect only 1% of the bare soils. More specifically, they make a large contribution to the metric to decametric specific drainage areas, and are responsible for half of the strongest signals exceeding $1 \text{ T/m}^2$.

As our study covers a period of about 6 years, it is reasonable to assume that these active unstable zones are not necessarily the result of a single slope failure, but may have experienced a succession of smaller movements, in particular to clear debris accumulated downstream by incision or headwater recession (Nadal-Romero and García-Ruiz, 2018). This is particularly the case for the 'La Coulée' landslide, which is still experiencing significant erosive activity two decades after it was triggered. Consequently, and despite the size of these active zones, we believe that our results correlate well with those of Wijdenes and 350 Ergenzinger (1998) and Yamakoshi et al. (2009) on miniature debris flows (MDF). These authors claim that they play a crucial role in the transport of coarse sediments, contributing between 5 and 36% of the total export of the neighbouring Roubine basin (Yamakoshi et al., 2009).

Finally, our study shows that an average of 29% of landslide deposits remain on the slopes during the period. Supplementing the time series with other campaigns at high spatial resolution, as well as with specific surveys at the event scale, would make 355 it possible to monitor the drainage of these materials, characterise the processes involved and, more generally, describe the evolution of these active areas.

## 5.4 Opportunities in a changing climate

Mediterranean environments are among the most affected by climate change, with a projected significant decrease in precipitation (except in winter for the southern French Alps), an increase in temperature and an increase in the frequency of paroxysmal 360 events (Giorgi and Lionello, 2008). It is therefore crucial to assess the impact of these changes on the future evolution of critical zone processes, particularly for vulnerable environments such as the badlands, which are a major contributor to sediment export (Copard et al., 2018). How their erosive dynamics, which are closely linked to those of vegetation (Gallart et al., 2013), will evolve with a decrease in winter weathering caused by cryoclastic forcing (Ariagno et al., 2022), and a decrease in summer



precipitation competing with an increase in the number and intensity of summer storms triggering landslides (Gariano and Guzzetti, 2016; Turkington et al., 2016), is still controversial (Hirschberg et al., 2021; Nadal-Romero et al., 2022).

However, the widespread availability of high-resolution data is paving the way for the development of geomorphological analysis tools capable of quantifying and spatialising sediment sources and sinks. The methodology developed in this study offers new prospects for analysing erosion and sediment transport at different scales within a catchment. It represents a promising complement to existing observation methods, which could help better constrain hydro-sedimentary transport models that already accurately simulate runoff response to precipitation forcing in these catchments, but which have a more limited predictive ability for sediment fluxes (Lukey et al., 2000; Mathys et al., 2003; Carriere et al., 2020; Bunel et al., 2025).

## 6 Conclusion

We combined a diachronic analysis of LiDAR data acquired at 6-year interval with a material bulk density model to analyse erosion in a small badland catchment. We were able to evaluate a total mass loss of $60\pm20$ kT, corresponding to an annual erosion of 200 $\mathrm{T.ha^{-1}.yr^{-1}}$ on denuded areas, which is 23% less than the export measured at the long-term outlet hydro-sedimentary station. We found that landslides and ridge failures are important contributors to the total flux (15% of the total flux for 1% of the total surface), and that the low drainage areas are the most productive (20% of the total erosion for 7% of the total surface). Our method appears to be a very promising approach for assessing sediment transport in badlands under a changing climate.

*Data availability.* Hydro-sedimentary chronicle data are available on the BDOH database repository (https://bdoh.inrae.fr/DRAIX/, Draix-Bleone Observatory (2015)). The LiDAR HD database is available online at https://geoservices.ign.fr/lidarhd (IGN, 2024).

*Author contributions.* YB drove the science, the LiDAR processing chain and produced the results. AL initiated the research. CL and SK provided data and expertise that support this research. GC 3D-printed the reglets used in the density measurements. YB, AL, CL, SJ contributed to writing the paper and sharing ideas.

*Competing interests.* The contact author has declared that none of the authors have any competing interests.

*Acknowledgements.* The authors acknowledge the support of the CNES (APR STERREO), the Programme National de Télédétecion Spatiale (PNTS, Grant PNTS-2022) and the LabEx UnivEarthS (ANR-10- LABX-0023 and ANR-18-IDEX-0001). This study was carried out at the Draix–Bléone Observatory (France), using infrastructure and data. The Draix–Bléone Observatory is funded by the INRAE (National Research Institute for Agriculture, Food and Environment), the INSU (National Institute of Sciences of the Universe) and the OSUG (Greno-





ble Observatory of Sciences of the Universe) and is part of OZCAR, the French network of critical zone observatories, which is supported
by the French Ministry of Research and French research institutes and universities.

**Appendix A: Calculation of local volume changes from cloud-to-cloud distances**

The M3C2 algorithm (Lague et al., 2013) is used to evaluate the local distances between two point clouds. A Geotiff raster
with a resolution of $1\ \mathrm{m}^2$ pixels is constructed using 5 scalar fields:

– the average distance between clouds in each cell $h_\perp$;

– the corresponding uncertainty value $dh_\perp$;

– the cell point population $p$;

– the mean point height $z$;

– the uncertainty on this height value $dz$.

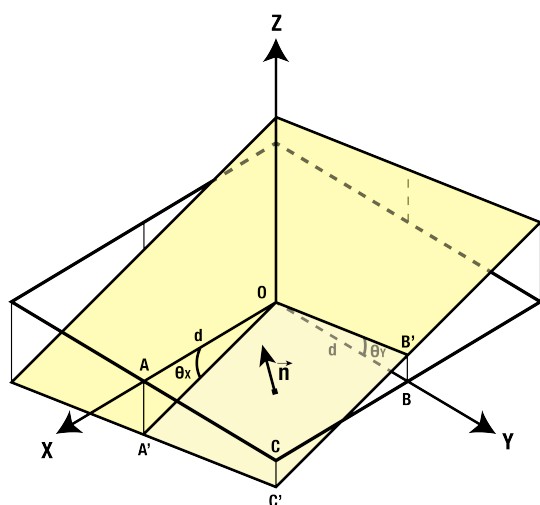

**Figure A1.** Geometry used to calculate the area of oriented facets describing local surface depletion or accumulation.

The surface model is then used to derive the gradient components in the X and Y directions of the grid. The population of each
cell is information that can be used to filter cells if, for example, an outlier value is suspected. To calculate the area intercepted
by each mesh in the grid, the grid is locally modelled by a plane whose inclination $(\theta_x, \theta_y)$ is given by the components of the
gradient in the X and Y directions. The same assumptions are made as for the shallow water equations (see Sect. 4.3), namely
that the choice of grid size causes the length scale for the curvature of the topography to be much larger than the length scale
at which normal variations $h_\perp$ are considered. Considering one quadrant of each facet and the plane equation:





$$
\begin{cases}
\text{OA'} = (d, 0, d \times \tan\theta_x) \\
\text{OB'} = (0, d, d \times \tan\theta_y)
\end{cases}
\implies \text{A'B'} = d \times (-1, 1, \tan\theta_y - \tan\theta_x)
$$

$\exists\, \boldsymbol{n} \,|\, \text{O', A', B', C'}$ satisfy $n_x x + n_y y + n_z z = 0$

$\implies x \tan\theta_x + y \tan\theta_y - z = 0$

$\implies \text{OC'} = d \times (1, 1, \tan\theta_x + \tan\theta_y)$

This gives the area $\mathcal{S}_{OA'B'C'}$ of each facet quadrant as half the product of the diagonals:

$$
\mathcal{S}_{OA'B'C'} = \frac{d^2}{2} \sqrt{(2 + (\tan\theta_x + \tan\theta_y)^2} \sqrt{(2 + (\tan\theta_x - \tan\theta_y)^2}
$$

Voxels can be formed in each cell: the third dimension is given by the M3C2 distance and their volume corresponds to the local erosion/deposition value:

$$
\mathcal{V}_{voxel} = 4 \times \mathcal{S}_{OA'B'C'} \times h_\perp
$$

Note that the sign of $\mathcal{V}_{voxel}$ is given by the sign convention for $h_\perp$, which allows us to distinguish erosion values from deposition values. Uncertainties can also be applied to the volume calculation:

$$
d\mathcal{V} = h_\perp \left( \partial_{\theta_x}\mathcal{S}\, d\theta_x + \partial_{\theta_y}\mathcal{S}\, d\theta_y \right) + S_{OA'B'C'}\, dh_\perp
$$

where $dh_\perp$ is one of the scalar fields contained in the Geotiff frame, as well as $dz_{x+1}$, $dz_{x-1} dz_{y-1}$, and $dz_{y+1}$, which allows $d\theta_x$ and $d\theta_y$ to be calculated. The barycentre of each cell itself has a positional uncertainty in the plane which is also taken into account by the cell population $p$.

## Appendix B: Construction of the bulk density model

### B1   Density of sediment deposits

In the deposition beach, the dry density value used in the data evaluation is 1.7 (Klotz et al., 2023). However, measurements carried out in 2001 in the Roubine trap, an elementary gully adjacent to the Laval basin, gave a lower average value of 1.5 (Mathys, 2006). In this work, density measurements were carried out at various sites in the Laval catchment, mainly in sediment deposits along the channel and its banks, and in colluvium at the bottom of slopes and gullies. The experimental protocol was as follows:

1.  Sampling a sediment deposit using a shovel and an 11 L bucket;



2. Weighing the sample using a hook balance;

3. Placing 3D-printed reglets around the sample area;

4. Photographing the sample area in different orientations;

5. Sampling a portion of the deposit using a vial;

6. Wet/dry weight measuring before/after 48h oven drying and determining of the initial water content of the sample;

7. Estimating the in situ volume of the sample by photogrammetry, based on the photographs taken in the field;

8. Calculating the dry density of the deposit from the measured mass, the occupied volume and the estimated water content $d_{dry} = d_{wet}/(1+\chi)$.

The measurement uncertainties include those related to the weight of the sample taken from the 11 L bucket, those related to the volume measurement and those related to the water content $\chi$. A value of 10% of the weight can be assumed, while the

volume uncertainty is typically 100 mL, i.e., about 2% of the volume. As a result, the wet density values are known within an average range of 0.2. The water content could not be determined for all samples. The values measured in March 2024 ranged from 11.96 to 15.65, with the corresponding dry density reduced from 86.5% to 89.3%. Where this water content was not measured, a value of 14±10% was assumed. This gives an average uncertainty on the dry density values of 0.25.

Density measurement campaigns were carried out in March 2024 and June 2024 in the sediment deposits of the Draix Laval

basin. In March, the water content of the samples was measured, while in June the mean value of 14% was used with an error of 10%. Table B1 shows the density measurements according to the geomorphological processes involved. It can be seen that scree slopes have particularly low densities, and that landslide or debris flows are also significantly less dense than deposits in the bed or in alluvial terraces.

**Table B1.** Density measurements according to geomorphological processes.

| Process | Dry density |
|---|---|
| Scree slope apron at the bottom of the slope | $1.14 \pm 0.4$ |
| Colluvium at the bottom of landslides | $1.41 \pm 0.6$ |
| Alluvial terrace (surface) | $1.69 \pm 0.8$ |
| Alluvial stock in the riverbed (surface) | $1.59 \pm 0.9$ |

A systematic survey of the densities in these different compartments, as well as a common measurement of the water content

of the samples taken, would allow this preliminary study to be carried out in greater depth. A mean value of 1.40±0.3 seems to cover the different types of sediments in the catchment with the same effective density.




## B2 Density of eroded materials

A study carried out over 2.5 years by Ariagno et al. (2023) in the Moulin basin, adjacent to the Laval basin, measured the effective densities and water content of the surface layer (0-2.5 cm deep) of marl subject to weathering and erosion, and determined how these change over the seasons. It appears that dry density is minimal in late winter (1.39±0.2) and early spring (1.55±0.2), when the slopes have developed an upper layer of weathering, and minimal in late summer (1.76±0.2) and autumn (1.71±0.3), when this layer has been washed by seasonal storms. The average value of 1.61±0.2 emerges. Mathys et al. (1996) used the value of 1.3 to describe the erosion rate of weathered marl, typically around $11 - 15 \mathrm{mm.yr}^{-1}$ at Laval, based on the ratio of sediment production at the outlet to the area of bare land. Using the value of 1.39 (Ariagno et al., 2023) at the end of winter, i.e., when the eroded material is made available on the slopes, this result can be re-evaluated at $10 - 14 \mathrm{mm.yr}^{-1}$. If we now consider the production values in Table 2, we find an equivalent ablation value for weathered marl of 18-20 $\mathrm{mm.yr}^{-1}$ at the surface, with strong inter-annual variation. As a result, we can estimate the expected average depth of denudation of the slopes in the 2015-2021 interseason at no more than 9-12 cm, i.e. the average thickness of this detrital layer on the slopes according to the study carried out by Maquaire et al. (2002). This means that between these two campaigns, the entire detrital layer present in 2015 has been purged. We choose 1.60±0.3 for the density of this surface layer. Locally, we can also expect the outermost layer of regolith, which was just weathered during the first campaign, to be ground down. It is somewhat harder and denser, but a priori less so than the less weathered or intact marl between 2.1 and 2.65 (Lavergne, 1986; Mathys et al., 1996; Serratrice, 2017). The above study by Maquaire et al. (2002) estimates this horizon of unaltered marl to be about 45 cm deep.

## B3 Marl density profiles

The considerations from the previous two subsections allow the construction of a simplified density profile model, constant with depth at 1.4±0.3 for sedimentary deposits and linear from 1.6±0.3 at the surface to 2.65±0.3 (not exceeding 2.8) at 45 cm (then constant). To incise a layer at a given depth, the upper layers must first be eroded. Thus, at a local distance $d$ measured between the local surfaces of two surveys, one is sensitive to an effective density $\rho_{eff}$ (Fig. 2), which takes into account the erosion of the overlying layers $M_{loc} = \rho_{eff} \times S \times d = \int_0^d \rho(z) \times S \times dz$. This is the physical quantity used in Sect. 3.4.

## Appendix C: Focus on uncertainties in hydro-sedimentary chronicle data

In Sect. 3.1 we present the cumulative sediment export of suspended and deposited sediments at the outlet of the watershed calculated with:

$$
\begin{cases}
M_{susp.} = \sum_t C_{MES}(t) \times Q(t) \\
M_{dep.} = \sum_t V_{deposit}(t) \times 1.7
\end{cases}
$$





In the data paper of Klotz et al. (2023), the characterisation of measurement uncertainties is presented in the form of quality codes assigned during the expertise of the data (Table C1). Without further assumptions on their distribution, they are considered as expanded uncertainties propagated by the following equations:

$$
\begin{cases}
\Delta \left( \sum_i X_i \right) = \sum_i \Delta(X_i) \\
\dfrac{\Delta \left( \prod_i X_i \right)}{\prod_i X_i} = \sum_i \dfrac{\Delta(X_i)}{X_i}
\end{cases}
$$

**Table C1.** Correlation between quality codes and uncertainties in hydro-sedimentary data, adapted from Klotz et al. (2023).

| Code | 1 : No quality attributed | 2 : Good quality | 3 : Intermediate quality |
|---|---|---|---|
| $Q$ (m$^3$.s$^{-1}$) | 30% (assumption of this study) | 10% (no issues noted) | 30% (flooded gauging system or deposit trap full of sediments) |
| $C_{susp.}$ (g.L$^{-1}$) | 60 % for $C_{susp.} < 50$g.L$^{-1}$ and 13% for $C_{susp.} > 50$g.L$^{-1}$ | 10% (event-scale calibration) | 60% for $C_{susp.} < 50$g.L$^{-1}$ and 13% for $C_{susp.} > 50$g.L$^{-1}$ (inter annual turbidity calibration) |
| $V_{depos.}$ (m$^3$) | 10% (subsidence of drying material) | | |



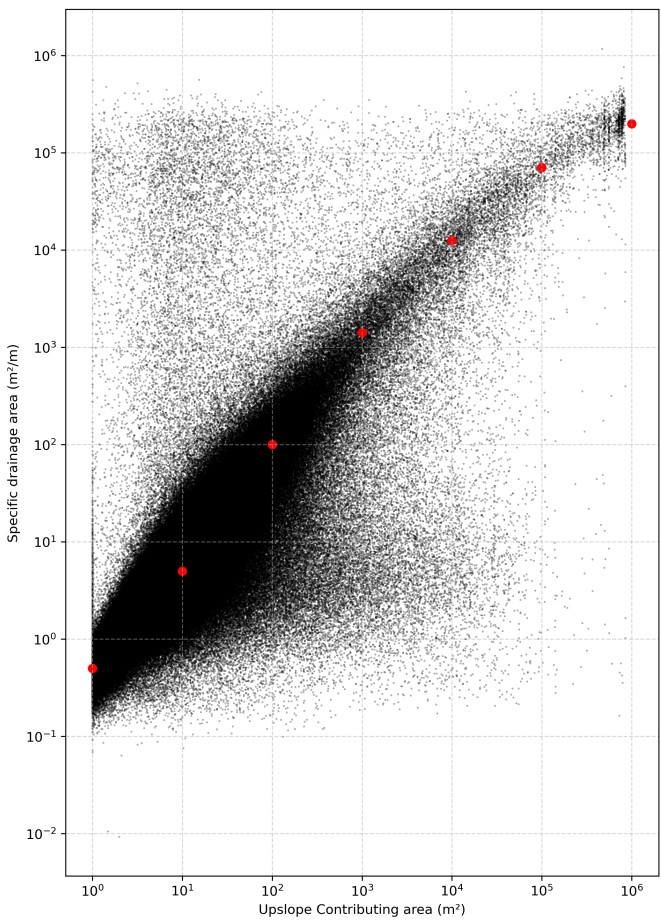

**Figure D1.** Specific drainage area $(\mathrm{m^2/m})$ against upslope contributing area $(\mathrm{m^2})$ for a runoff rate of $50\,\mathrm{mm/h}$.

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





**Figure E1.** a) Location of the study areas used to characterise the effect of a centimetric co-registration error between two campaigns. Background: Aerial photograph of the outlet of the Draix-Laval basin (IGN, 2021). b) Distribution of local distances between ground points of the 2021 campaign and the 2015 campaign on the study areas with or without a $(\Delta X, \Delta Y, \Delta Z) = (10, 11, 0.5)$ corrective shift of the 2021 point cloud.