# Peer review of "Spatial assessment of erosive processes in a badland catchment using diachronic LiDAR, Draix, Alpes de Haute-Provence, France"

_EGUsphere, 2025_

## Author Response (AR1)

---- ----

We sincerely thank the reviewers for their thoughtful and constructive feedback. All suggestions have been carefully addressed, and detailed responses to each comment are provided below (reply to RC2 begins on p13). At the end of the document, a table summarising the changes made to the structure of the manuscript is also included, as well as a list of the new references that we included to the paper (p19-20).

The main comments were as follows:

- to improve the clarity of the manuscript by reworking certain formulations, adapting the terminology used, adding illustrations, and reorganizing the article's structure. These changes have been made.
- to further discuss the geomorphological processes involved, taking into account other morphometric analysis studies. These suggestions have been addressed, particularly in the Discussion section, based on the references provided and by incorporating additional ones listed at the end of the document. Although such analyses do not constitute the core originality of our study, we considered that a more detailed examination of the geomorphological processes would be a natural perspective for future work.

---- Response to reviewers ----

**RC1**:**

**Global remarks:**

"I believe the specific research questions addressed in this manuscript could be set out more clearly."

To address this, we have adjusted the title formulation to make explicit that the developed method, by providing a precise and spatially distributed quantification of topographic mass changes over a given period, allows us to tackle the question of in situ sediment production within the catchment. Similarly, the abstract now includes the notion of "fingerprinting soil losses." This point is also reiterated at the end of the introduction, which has been slightly revised, as detailed in response to a later comment.

"I also found that the methods were described with a variable level of detail and that whilst individual errors were outlined well (Section 5.1), it was not clear to me how these errors were incorporated into the total sediment budget."

To better clarify the method—as will be detailed further in response to several specific comments—we have restructured this section so that the various components leading to the presented results are placed on an equal hierarchical level within the paper. These components are now presented in an order more consistent with both the site and data presentation in Section 2 and the results presented in Section 4. As mentioned later, the paragraph reconstructing the hydrological network has also been moved into the Methods section, and explanations that were initially included in the Appendix have been reintegrated into the main text. A diagram has been added to illustrate the LiDAR workflow (Figure 3), which consists of three steps, now also presented at the same hierarchical level in the paper. A

summary table of the article's structural changes is provided at the end of this document. Regarding uncertainty, while it is discussed in Section 5.1 (now 5.5), its introduction in earlier sections—particularly in Section 3 (Methods) and Section 4 (Results)—lacked clarity. We have therefore clarified, at each computational step within the Methods section, how the associated uncertainties are calculated. Additionally, when discussing the overall sediment budget in the Results section (notably in Subsection 4.2), we have explicitly detailed the origin and computation of uncertainties, with the aim of ensuring greater transparency. "There were quite a lot of different terms used within the manuscript, which I think would benefit from definitions earlier on. I believe with some moderate changes, the authors can

As explained below, the original text made excessive use of varied terminology to refer to concepts that were sometimes similar, and occasionally employed uncommon terms in geomorphology and remote sensing without providing clear definitions. We have undertaken a homogenization of the article's terminology by replacing uncommon or imprecise terms and by explicitly defining the remaining terms upon their first introduction.

improve the quality and readability of the manuscript."

**Clarity of the paper:**

« The introduction presents an overview of recent studies that explore different processes in badland landscapes on a variety of different scales. Whilst the authors briefly mention a wide range of literature, I wonder if the structure of the introduction could be reworked to more closely align with the work conducted here. For example, it would be helpful to know where the processes described in the small-scale studies are thought to occur within the landscape (lines 22 - 35). »

In an effort to revise the text in a way that is both minimal and meaningful, while also addressing this comment, we have reworked the introduction to better clarify the reasoning that highlights the relevance of a quantitative study based on high spatial resolution data to identify erosion sources within this catchment. We discuss the contributions of conventional methods used to characterize erosion and its processes—for instance, plot-scale studies on slopes or outlet-based measurements that integrate erosive and hydrological activity at the catchment scale. We then outline how very high-resolution remote sensing, which has already helped overcome key scientific limitations, can further advance our understanding of the spatial distribution and quantification of erosion, particularly when the entire catchment is covered using airborne or drone-based methods.

« I think the manuscript would also benefit from a set of definitions near the start of the manuscript. This will help to develop a narrative that can be used to set out how the sediment budget was calculated and will ensure the research is accessible to different fields of geomorphology. Terms that I personally would appreciate being defined in this context are: mass balance, sediment production, critical zone compartments, classes of sediment sinks/ stores (these may benefit from a schematic figure). For some of these definitions, specifying the units may also be helpful. »

As mentioned in response to the general comments, and in order to address this specific point, we have favored more commonly used terminology—for example, consistently using "sediment budget" instead of the previously interchangeable use with "catchment-scale mass balance," now clearly defined as such at the end of the introduction (L77–79). The same applies to "sediment production" (L74-75). The term "critical zone compartments" has been

replaced with "geomorphological compartments," although a definition of the "critical zone" is still provided when it first appears in Section 2.1.2. The introduction of sediment "sources" and "sinks" has been revised in Sections 3.4 (Methods) and 4.1 (Results) to ensure greater clarity.

More generally, we avoided the use of terms such as "decades" (replaced with logarithmic intervals), "diachronic" (replaced with "multi-temporal"), "paroxysmal" (now described as "intense" or "extreme", with clarification of the intended meaning), "accumulation beach" (more accurately described as a sediment trap), and "modes" (which lacked clarity and has generally been replaced with "processes"). The term "scour" was replaced with "erosion," and "chronicles" with "records."

Synonyms were systematically avoided throughout the text to ensure consistency, and only terminology or nuances specifically chosen by cited authors were retained. For instance, "fluvial domain" was preferred over "hydrographic network compartment," and this correspondence is explicitly clarified (L364).

**Methods**

« The manuscript would benefit from an additional workflow figure that shows how the different methods used in this paper are connected (it could also link to the definitions suggested above). There are an impressive number of high-resolution datasets and analyses conducted in this paper, however I often lost sight of the different approaches, and in particular over which compartment of the landscape they were used. A workflow that connects each approach as well as describes the temporal intervals (e.g. 6-year differencing on point clouds or 1 year on the sediment trap), spatial scale and errors (see next point) would be helpful for the reader to appreciate the breadth of this study. »

To address this comment, we primarily opted for a more balanced reorganization of the Methods section, aiming to clarify its structure and to better articulate the presentation of the calculations and measurements performed. This involved, as proposed, moving part of the former Section 4.3—originally placed in the Results—into the Methods (now Section 3.4), as well as integrating the content from the originally designated Appendix C into the main text. These elements have been redistributed between Section 2.3 (for density measurements) and Section 3.2 (for the construction of the density model).

Additionally, the internal hierarchy of the Methods section was adjusted to place each step at the same structural level, in line with Sections 2 (Data Presentation) and 4 (Results), highlighting each step as a key component of the overall methodological framework. This improved organization is announced in the final paragraph of the introduction (L74–84) and reiterated in the opening of Section 3.

Finally, a workflow diagram (Figure 3) has been added to the article to illustrate the three steps used to reconstruct local volumetric changes in the catchment based on point clouds from the LiDAR surveys (Section 3.1). We have thus responded to the underlying concern of this comment, which seemed to point to a lack of clarity in the structure of the Methods section. However, we deliberately chose to include a workflow diagram only for Subsection 3.1, which, in our view, lends itself best to such a visual representation and significantly enhances understanding of this core component of the method.

« I appreciated the clarity of Section 5.1 which described the potential errors for the methods used. However, I found it unclear how these values were consolidated into a single value and applied to the overall mass balance. Please can you add additional details, this could be within the workflow. »

Thank you for this comment. We indeed aimed to emphasize the evaluation of uncertainties and methodological constraints by dedicating a specific section to this topic within the Discussion (Section 5). To ensure greater clarity in the calculation of these errors, we have included a statement on the associated uncertainties and their propagation in each subsection of the Methods section. Furthermore, we have completely reworded the first paragraph of Section 4.2 to clarify that the expanded uncertainties in the sum of local mass changes predominantly stem from the lower- and upperbounds applied to the density profiles, rather than from the propagation of errors related to topographic reconstruction and volumetric changes. This issue is discussed in greater detail in Section 5.1 (now 5.5). Additionally, we have revised the wording in the section presenting the density model (Section 3.2) to clearly explain that lower and upper bounds are defined for the relationship between erosion/deposition depth and bulk density.

« Integration of Appendix B and C into the main text. In my opinion the steps taken to measure bulk density and calculate the uncertainties on sediment export are important for the readers overall understanding of the analyses conducted here. I have provided more detailed comments on these sections below but think that a refined version of these can be integrated into the main text. »

As part of the reorganization of the Methods section mentioned in previous responses, we followed the recommendation to integrate Appendix C into the main body of the article and to separate the presentation of density measurements (Section 2.3) from the construction of the density model (Section 3.2). Appendix B was only marginally revised, and the reference to this appendix within the main text (Section 3.1, now Section 3.3) was also reformulated to make it less essential for understanding that paragraph.

« Similarly, the steps taken to generate the hydrographic network may be better placed in the methods, as first mentioning this in the results was unexpected. » Done, as mentioned above.

**Figures**

« I would recommend several new figures to improve the clarity of the research conducted. »

« ... Methods workflow (see above)... »

Done for the LiDAR workflow, but we also address the corresponding overall comment (cited previously) through a comprehensive restructuring of the Methods section and a reformulation of its introduction.

« Sediment budget schematic with values and units:

The mass balance is presented using many different metrics. Whilst this is great because it shows the volume of data and scales over which different sediment contributions can be quantified, I think the overall conclusions of the paper would be clearer with a diagram that shows the different critical zone compartments, the different processes therein and the estimated mass balance for each process. This could also be a good opportunity to show how

these zones link, i.e. a more connected version of Fig. 5, for example a Sankey diagram? Or a schematic showing the different stores.

A schematic showing how transport vs supply limited conditions could influence the point cloud differencing. This is currently summarised in lines 260 to 270. »

These two proposals were addressed together and appear to significantly enhance Section 4.4 (now 5.1), which calculates sediment production for the geomorphological compartments constructed from the hydrographic network. They took the form of the diagram in Figure 8, which visually represents the different compartments—illustrating their vertical arrangement, the sediment production calculated in this paragraph, as well as the geomorphological processes potentially involved. Additionally, two insets were added concerning two masswasting events whose source regions are located either in the crests or slope compartments, with deposits in the slope compartment. A paragraph was also added to Section 4.4 explaining, through these examples, how the calculation method, following the observational sequence, attributes the sediment production of these mass movements to one compartment or another, depending on erosion regimes limited either by transport or by sediment supply.

« Discrepancy between the outlet and sediment generated on hillslopes. Whilst the disconnect between the sediment at the outlet and the sediment eroded (exports more than is eroded) is acknowledged, there is little explanation for this. I think the manuscript would benefit from a greater discussion of this discrepancy (beyond line 190). »

The explanation of this point lacked clarity, so we have reformulated the beginning of Section 4.2 to better clarify the role attributed to uncertainties in this discrepancy. Indeed, the confidence intervals of the mass balance values and the measured export overlap, and the sources of uncertainty are significant. It is precisely to explain this gap that we initially chose to dedicate an entire section to the uncertainties and limitations of the method (Section 5.1, now 5.5).

**Unusual phrasing**

A few times within the manuscript, unusual phrases are used. I have highlighted my main concerns below.

"diachronic" – "from a quick search this term appears infrequently in the literature, I would suggest multi-temporal or a similar phrase as this is more commonly used to refer to landslide inventories that use more than one time interval." Systematically replaced by 'multi-temporal'.

"paroxysmal" – replace with extreme? Replaced by « extreme » (L95)

"decade" to describe the order of magnitude increases in specific drainage area. I would recommend changing this phrase as it may be assumed that you are referring to a temporal scale. Systematically replaced by 'logarithmic intervals' or 'log intervals'.

A thorough read through of each sentence is necessary before resubmitting. Common errors I noticed included:

- Use of the word "some" or equivalent, can you provide more specific values. We arranged that.
- a) not consistent with journal formatting. Modified
- Ensure you have spaces between values and units. Checked
- Use either / or superscript. We chose superscripts.

**Line by Line comments**

**Abstract**

Line 1: 250 T ha-1 yr-1 - I am not sure where this value is from as it is not in the introduction. Actually, we have slightly modified the approach and included a reference in the introduction (L23) citing Mathys (1996). The value is recalculated within this study but expressed in kg·m-2·yr-1, as discussed in Section 5.1 when addressing production rates. In the original units—commonly used for sediment export—we obtain an average of 210 T·ha-1·yr-1 at the catchment scale over the studied period, with crests actually rising to 420 T·ha-1·yr-1. This is why we retained the general statement of "more than 200 T·ha-1·yr-1." This value is sometimes also expressed as "centimeters of fresh/weathered marl per year."

Line 4: replace chronicles with reports

We preferred to replace it with "(hydro-sedimentary) records," which we felt might be even more appropriate.

Line 5: diachronic to multi-temporal? See above. Done

Line 6: "modelling" to "model". Done

Line 6: remove "out". Done.

Line 7: represent not represents. Done

Line 8: Sentence starting "They contribute..." is unclear, please can the authors rephrase. Done (L9)

Line 10: "seams" to "seems". Done

Line 10: "quantifying and localising" unsure about the use of these terms here. Could rephrase to something along the lines of "our approach is promising when identifying local erosion hot spots and quantifying their contribution..." Done based on your suggestion (L10-12)

**Introduction**

Line 14: Landforms to landscape? Done

Line 17: Use numeric values as opposed to roman numerals. Done

Line 20: Could you provide an estimate of spatial dimensions for plot scale for those unfamiliar with this? Done

Line 26: The sentence "It should be noted..." is unclear, please rephrase. Done (L31)

Line 43/44: What does annual export values refer to here? We have made an effort to harmonize the use of terms such as sediment production, annual export values, and sediment export throughout the rest of the text. Here, the terminology follows Nadal-Romero et al. (2011) to designate annual sediment export expressed in T·ha-1·yr-1. Their study compiles data from different types of measurements and from drainage areas of highly variable sizes, ranging from submetric slope segments to catchments of high Strahler order.

Line 45: is this a power-law decrease in sediment production? In sediment export, we have further specified the term.

Line 47: a to an. Done

Line 50: "This calls for", I like that the authors have summarised the research gap, however I think this could be more specific. For example, what specifically calls for a multi-scale study? We reworked a little the sentence to specify it (L57-59)

Line 60: "corresponding mobilised masses" – I'm unsure what this refers to? Clarified.

Line 61: Could you provide more insight into why there may be variations in compaction in these landscapes. The sentence was slightly rephrased (L69–71); it refers to a variation in the porosity of the material related to the destructuring of fresh marls.

Study site and data

Line 83: paroxysmal could change to extreme (see above). Done

Line 83: 800 g/L – is this suspended sediment concentration, please clarify. We corrected the phrasing to "several hundred grams per litre," as specifying an exact value was not very meaningful. We clarified that we are referring to suspended sediment concentration. The data paper by Klotz (2023), cited in the article, establishes this relationship:

Figure 1: I think Figure 1 could be used to provide a more effective overview of the catchment. For example, is it possible to add vegetation cover for the Laval catchment to the plot so that the readers can see the area over which the mass balance has been calculated. Additional panels with field photos showing the processes observed would also be a nice addition. The caption should be expanded to explain the shading on the plots and describe each panel in turn. The location of the Laval basin should be shown in Panel A. Please label each panel. We have incorporated all these comments.

Line 94: Could you expand on a Parshall flume? The sentence in which it was mentioned has been rephrased (L108–110) to clarify that it refers to the device used to measure flow rate through water level measurement combined with knowledge of the Parshall flume geometry. This is illustrated in Figure 1.

Line 95: Do you have a grain size limit for the coarsest materials? At line 112, we added more information on the measured grain sizes, notably the D90, which provides an overview of the dimensions of the coarsest materials measured in suspension.

Line 96: Could you add photographs of the sediment trap to Figure 1 or the recommended workflow. Do you have a record of how many intense events occurred within the catchment between 2015 and 2021. That may be interesting for context. A photo of the sediment trap has been added to Figure 1. An additional figure (Figure 2) presents the precipitation records, including intensity and monthly cumulative values over the study period. This figure also contextualizes the dates of the two measurement campaigns. A sentence at the end of this paragraph (L116–117) summarizes the number of intense events during the period.

The use of point clouds appears robust, however this is not an area which I have expertise in, so hopefully the other reviewer can provide a more thorough assessment. We did not receive specific comments regarding the point clouds, except for clarifying over how many control surfaces the uncertainties of the first campaign's point cloud were measured.

**Methods**

Line 120: Chronicles to reports. We preferred "records," which we found even more appropriate.

Line 121: "instantaneous discharge" – what does this refer to? When are discharge measurements taken and which are used. This could be included in a methods workflow. We removed "instantaneous" and specified at line 110 that the flow rate is measured at a 10-second frequency.

Line 124: What does "It" refer to at the start of this sentence? To the cumulated export, we rephrased.

Line 125: "in the same way" is quite informal. Replaced by "Similarly"

Table 2: I like this table, it is nice to see the differences between the years. If this is the first time, Msusp, Mdep and Mtot are used, it would be good to define these. The authors could also add in the caption how the errors are calculated for each year. M susp, M dep, and

M\_tot are defined in the table legend; M\_tot, which is subsequently used, is also explained in the text. A reference is made to Appendix B, which details the calculation and propagation of uncertainties. It is also mentioned in the text that these are expanded uncertainties.

Line 150: I found this description unclear "This gives an average value to the different fields" Rephrased (L198-199)

Line 158: "induce" to "infer" though I think the sentence would benefit from being rephrased. The sentence has been slightly rephrased (L206–209) using the term "implies."

Line 167: "local distance" this may be common terminology to use when differencing a point cloud but I was not sure. By restructuring this section, the presentation of the local distance calculation is now included in Section 3.1.2.

I think it would be useful to add Appendices B and C into the main text. I also think it would be helpful to introduce the hydrographic network modelling in this section too.

As explained in response to the general comments concerning the Methods section, the structure of this section has been revised to incorporate the construction of the density model from Appendix C into a dedicated subsection 3.2; the measurement part has been moved to a dedicated subsection 2.3. Appendix B is retained but has been revised to provide more precise explanations of the calculations. In subsection 3.3, which refers to Appendix B, sentence structures have been reformulated to convey essential information, while details are referred to the appendix and the data paper by Klotz (2023).

Could you expand on the landslide mapping – are these mapped from orthophotographs? How are the scars and deposits distinguished, as I noticed there is quite a bit of disconnect between landslide scars and deposits, which is interesting. You could expand on the identification for each process within the sediment source and sink class so that when they are first referred to as classes in the Results (line 204), the readers are not wondering what these classes are.

By moving the introduction of the hydrographic network reconstruction to the Methods section (subsection 3.4, L240–244), we also introduced for the first time the mapping of erosion hotspots, notably landslides, mentioning that this was performed manually (using GIS software) based on Figures 5 and 6, and utilizing topographic information (DEM, slope) as well as an orthophoto. This is further detailed in the following Section 4.1, where these two figures are presented (L269–275).

**Results**

The results section could begin with a few statements about the spatial patterns of landslides and crests, e.g. 29% of deposits remained on hillslopes. This will mean my comment at Line 188 will no longer be relevant. Could also include XX% of landslides were on slopes etc. Sections 3.4 and 4.1 demonstrate how erosion hotspots are identified and associated with mass movements where erosion and deposition areas are visible. In Section 4.2, we discuss, based on the calculated mass of these deposits, the proportion they represent relative to the erosion zones. It is determined that 29% of the deposits remain on the slopes. A reformulation effort has been undertaken.

Fig 3. I really like this figure. Could you add a description of the flow direction (either in the figure or caption). We added a label for the outlet station on Figure 5 and 6.

Line 180: Use of "some strong signals" here is a bit vague, could you be more specific? We clarified the term "above  $\pm$  1 T.m2" to be more precise and to correspond with the discussion at L265–266.

Line 181: What is the main drain? A labelled schematic may help here to differentiate between the sediment trap and the main drain. We intended to refer to the main channel and have updated the text accordingly.

Line 188: I missed where the 71% used to assume landslide connectivity is from. We have sought to clarify the labeling of sources in Sections 3.4 and 4.1 to make it clear that deposits are associated with erosion zones for landslides, and that the quantification of local mass variations allows us to determine the proportion of landslide-related deposits remaining on the slopes. See also the following remark.

Line 189: "drained on average" I'm not sure about this phrasing. We reformulated the sentence to clarify that we mean 29% of the mass displaced by the landslide remains on the slopes (L287).

Line 194: Please reframe the sentence "According to Malet (2003)..." done.

Line 196: "This dynamic continues", I don't think this is the correct use of dynamic. Are you suggesting that the large volumes of sediment recorded at the output could relate to the reservoir caused by the large landslides? Interesting! We reformulated this passage (L294) to clarify that although a major landslide obstructed the main channel in 1998, topographic reworking at this location is still observed more than 20 years later, notably resulting in a continuous sediment supply to the outlet.

Line 207: Unsure about the use of "about" here, was the input value 50 mm/h? If so, you can remove "about". removed

Figure 4: You could add the resolution of the DEM used in the caption. Replace the term "decades" (see earlier comment). done

Line 217: Figure 5.a) replace with Figure 5a. done

Line 221: Delete "as already mentioned" replaced by « as mentioned in Sect. 4.1 » to be more specific

Line 228: "...it is possible to determine *their weight* in the sediment balance" I am confused by what you are calculating the weight of? This is also the first, and only, use of the term sediment balance. I think the manuscript will be much clearer by a defined set of terminology introduced in the introduction, as explained earlier. It was confusing and a poor choice of word, we meant "contribution" (L314).

Figure 5: Add panel labels – text refers to A and B. done

Line 230: I found the discussion about floodplains and levees with low drainage orders a bit confusing. Would it be possible to include a schematic in an appendix or supplemental that explains the process behind this? Perhaps the discussion on levees and floodplains could be presented as a paragraph at the end of 4.3. We chose to slightly rephrase lines 313–319 to enhance clarity. This discussion is illustrated in the diagram presented in the following Section 5.1.

Line 234: Similar comment as above about classes. Critical zone compartments are only mentioned three times in the manuscript but sound very important from the abstract. Could you expand on the definition of these and the different compartments. We changed the terminology to the more commonly used "geomorphological compartments," which are subsequently defined by the drained area (L316–342). A diagram will illustrate these compartments, and Section 5.2 discusses their delineation, particularly in relation to an alternative definition approach (L363–371) and Appendix C.b.

Line 240: Should ""Slopes" just be "slopes" similar to crests? corrected

Line 242: "mass balances" I think these terms need to be defined earlier. Replaced by "sediment budget" with a definition provided in the Introduction (L78–79), in the Methods section (L208–211), and when the calculation result is presented (L279–280) in the Results section 4.2

Line 248: "above that" sounds quite informal. Replaced by « above »

Line 248: "with the considerations" I'm not sure if considerations it the right word here. Replaced by "assumptions"

Line 252: What is the "accumulation beach" can this be added to a schematic? We replaced it by « sediment trap »

Line 256: "we should also take into account..." I found this quite vague, perhaps by adding the specific paragraph about floodplains and levees as suggested above, this can be reworded to be clearer. Rephrased (L373)

Line 269: "we are probably closer to the former case", could you explain why this assumption is made. The whole paragraph has been reworked (L407-410)

Line 278: "Although our framework..." I was a bit confused about this, so are all the estimates of mass balance from the crest or slopes? Are values from channels and gullies not included in the total? Please can the authors clarify. We proposed a bounding estimate for production within the hydrographic network. Consequently, this sentence has been replaced by the passages from lines L410 to L416.

Discussion

Line 293: Sentence starting "The design..." should be reworded. Done

Line 297: What does "a density shift" refer to here? rephrased

Lie 306: "shrubs" corrected

I think the spatial and temporal scales section will benefit from the schematic suggested earlier. This was done to illustrate transport-limited and supply-limited erosion regimes in the schematic presented in Figure 8.

Line 318: Unsure about "emergence". Corrected by "initiation"

Line 343: "metric to decametric" rephrase. Done "ranging from metres to tens of metres"

Line 369: "It represents a promising complement" to "Our methodology complements..."

Line 377: Fluxes are only used for the first time here. It would help if terminology is consistent throughout. Replaced by « export »

**Appendix B**

Line 424: what does "deposition beach" refer to here? Corrected by "sediment trap"

Line 429: Could you locate the sample locations for bulk density measurements on Figure 1? Unfortunately, we did not retain the precise locations, but these measurements were conducted in or near the channel, on its banks, in colluvium at the base of slopes and gullies, as well as in the deposition area of the "La Coulée" landslide.

Line 442: What are these values? These were water contents expressed as percentages; the "%" symbol was inadvertently omitted.

Line 465: unsure about the use of "purged" this sentence was removed when the appendix was integrated to the main text.

**Appendix C**

I found this section and Table C1 confusing. Perhaps a more in depth caption could explain the different headings – for example, why are no quality attributed and intermediate quality given the same percentages? The table caption has been completed, and an explanation is now provided between lines 568 and 571: « When no quality codes are attributed we assume an intermediate quality, as at that time poor quality data where classified as missing data. There are no data entries flagged as 'low quality' in these datasets, although this may occur for rainfall data for instance. »

**RC2**:**

**General comments**

"However, some moderate revisions could improve the overall structure and clarity of the manuscript. The organization would benefit from aligning each methods subsection with a corresponding results subsection. Additionally, certain paragraphs should be relocated to more appropriate sections (see specific comments)."

The overall structure of the manuscript has been revised to improve clarity. A comparison table provided in the appendix of this document outlines the changes made. In particular, density measurements have been added to the "Site and Data" section (Section 2.3), while the construction of the density model—previously described in Appendix C—is now detailed in the Methods section (Section 3.2). Since the core and starting point of the method is the reconstruction of local volume changes from LiDAR point clouds, this part—previously Sections 3.2 and 3.3—has been moved to the beginning of the Methods section and is now organized into three subsections reflecting the three main steps, with an explanatory workflow diagram added (Figure 3). The reconstruction of the hydrographic network has also been moved into the Methods section (Section 3.4), which now includes the initial description of how erosion hotspots are identified. The cumulative export at the outlet is now only used in the Results section (Section 4.2), after presenting the erosion map in Figure 5 (formerly Figure 3) in Section 4.1. Consequently, this export aspect is introduced in the Methods section after the LiDAR workflow that reconstructs volumetric variations and converts them into mass changes (Section 3.3), rather than at the beginning (as in the former Section 3.1). Similarly, the Discussion section has been restructured: it now opens with the interpretation of results (Sections 5.1 to 5.3), followed by an assessment of methodological limitations and perspectives (Sections 5.4 and 5.5).

The complexity of the geomorphic processes acting in the badland area is only partially addressed. The diachronic change detection mainly focuses on mass-wasting processes, while water erosion processes are only briefly mentioned and inferred from drainage area classifications of local mass variations. In my opinion, greater effort should be devoted to better understanding the effects of sheet, rill, and gully erosion, which seem to be highly prevalent in the area.

Finally, some previous studies on morphometric analyses on non-arid badland sites should be considered.

Consistent with previous findings (Marsico, 2021; Nadal-Romero et al., 2011; Mathys, 1996), our study shows that the smallest drained areas (i.e., crests and slopes) are the most active in the initial mobilization of weathered material. In contrast, the hydrographic network is primarily characterized by sediment re-mobilization processes, involving materials that have been previously made available in upstream compartments through mechanisms such as crest failures, landslides, and sheet erosion. While identifying and characterizing geomorphic processes is not the core objective of our method, this perspective is now more explicitly addressed in Sections 5.1 and 5.3. Nonetheless, we have expanded the discussion on geomorphic processes in Sections 4.3, 5.1, and 5.3, and integrated these elements into a new schematic illustration (Figure 8). The definition of geomorphological compartments has also been revised in Sections 4.3 and 5.2. We now compare our approach to the method based on inflection points in the Hydraulic Slope–Specific Drainage Area relationship, as originally proposed by Montgomery and Foufoula-Georgiou (1993) and adapted by Bernard et al. (2022). This comparison is addressed in response to a specific comment (L366–373).

**Introduction**

Since this site is a humid badland, please provide a brief description of its origin. If vegetation is not limited by climate, is human activity— even if ancient—an important factor?

This description has been added at the beginning of the introduction (L17–24), in order to better contextualize the study and its objectives.

Study area

It would be helpful to to elaborate a bit further about the average monthly rainfall intensity along with the average number of rainy days, and to briefly discuss when the main hillslope denudation processes occur throughout the year.

This description is now further developed in Section 2.1.1 (L92–98), in order to provide additional context and clarify the local geomorphological setting.

This description has also been supported by an additional figure (Figure 2), and a clarifying sentence has been added at the end of Section 2.1.2 (L115)

Fig. 1 could be improved: please, indicate the Bouinenc and Bléone rivers, as the drainage network organization and the positions of the described confluences are not entirely clear. Furthermore, could you include illustrative photos of the main landforms (e.g., landslides, crest collapses, gullies, etc.)? Done

**Method**

Did you calculate a Level of Detection, below which change detection is unreliable? How was this error propagated in the analysis? We do not perform change detection in the traditional sense. For the sediment budget—calculated by summing the mass variations across the catchment—we retain all data points, including those with low change values. However, to minimize potential errors, we perform an a priori refinement of the co-registration process (Section 3.1.1), which includes a reevaluation of uncertainties related to both the co-registration and the accuracy of the point clouds (L79–80). These uncertainties are propagated throughout the workflow, but they are estimated to be of second-order importance relative to those associated with the density model and the depth of erosion/deposition.

For the manual labelling of erosion hotspots, we considered features showing mass variations greater than  $\pm 1 \, \text{T} \cdot \text{m}^{-2}$ . To improve clarity regarding uncertainty propagation, we have revised relevant passages in the methods sections, including the beginning of Section 4.2 (which presents uncertainty ranges for the sediment budget), and elaborated further in the dedicated Section 5.3 (formerly Section 5.1).

Was the UAV helicopter equipped with the RTk instruments? How many control point were used?

No RTK instruments are mentioned in the campaign report; however, 30 GPS points were collected on two control surfaces and used for quality assessment.

**Results**

Fig. 3: What about landforms associated with water erosion? We clarified the introduction of erosion hotspot labelling in Sections 3.4 and 4.1. We identify landslides on slopes and crests, as well as sediment reservoirs within the hydrographic network. In the discussion (Section 5.4), we took up the idea—originally suggested as a possible perspective—of more detailed morphometric analyses, although this is not the main focus of the present article. The labelling presented in Section 4.1 represents a first approach, allowing us to assess sediment production from landslides, which appears to be significant and helps explain the high sediment production rates observed in areas with the smallest drainage areas. It appears that the processes observed by other authors within the hydrographic network contribute more to the remobilization of sediments produced upstream than to net sediment production via incision. Some evidence for this can be seen in Figure 3 (now Figure 5), but Section 5.3 (L447–457) further explains that observing transport processes within gullies would require a similar multi-temporal analysis at a seasonal timescale (Bechet, 2016; Ariagno, 2022). The responses to the following comments further complement this one.

Section 4.2 needs clarifications. First, what do you mean by "erosion modes"? Perhaps you intended to say "erosion processes"? Also, the term "scour" seems unclear—please clarify or consider using "erosion" instead.

This section, particularly its first paragraph, has been revised and the terminology has been replaced according to the suggestions.

Moreover, please clearly distinguish throughout the paragraph when you are referring to the mass balance measured at the outlet versus that derived from change detection based on high-resolution surveys. Done

Section 4.3: here you introduce a new method and analysis, not earlier described. Lines 206 -217 should be moved to the method section. Done

Regarding the process domain threshold detection in the drainage area values, I recommend consulting the paper by Vergari et al. "The use of the slope—area function to analyse process domains in complex badland landscapes" (Catena).

As already mentioned, water erosion processes seem to be inferred from the specific drainage area intervals. These processes should be analyzed in more detail, as they are predominant in badland areas. Considering the slope gradient values, in conjuction with the drainage area, could support the interpretation of transitions from sheet to gully erosion.

We have moved the calculation of sediment production for the identified geomorphological compartments to Section 5.2. This section now begins with a discussion (L364–371) demonstrating that the identification of these compartments is consistent with the method of Montgomery and Foufoula-Georgiou (1993), which underlies the study cited by Vergari et al. (2019), now referenced here (L371–373) for its application to badlands. To verify this, we used the version adapted by Bernard (2022), which replaces slope and drainage area with hydraulic slope and specific drainage area—two metrics computed by the Graphflood algorithm (previously mentioned in Section 4.3 and now in Section 3.4). The correspondence between these two approaches is now illustrated in Appendix C.b.

For landform identification from multitemporal surveys, the paper by Llena et al., "Geomorphic process signatures reshaping subhumid Mediterranean badlands: 1. Methodological development based on high-resolution topography" (ESPL), may provide valuable methodological insights and should also be considered in the Discussion.

We chose to perform a simple manual identification of the landforms discussed in Sections 4 and 5. However, the method developed by Llena (2020) appears to be a promising avenue for refining

landform characterization and is mentioned as a future perspective in Section 5.4. The terminology from this paper has sometimes been adopted in Sections 4 and 5, notably in the schematic of Figure 8, to designate the geomorphological processes that may mobilize (or remobilize) sediments across the various identified compartments.

Section 4.4 reads more like a discussion section and and should be moved accordingly. Done

DISCUSSION: Please revise this section in light of the above comments. Done. The Discussion section has been reorganized and expanded based on the reviewers' comments, particularly in Sections 5.1, 5.3, and 5.4, to better outline the limitations and perspectives of our results. Section 5.2 (formerly 4.4) has been developed to provide a clearer description of the approach leading to the calculation of production rates for the different geomorphological compartments, to discuss their segmentation in light of the method proposed by Vergari et al. (2019), and to present a summary schematic highlighting the geomorphological processes potentially active within these compartments, based on Llena (2020) and other references cited throughout the paper (notably in the Introduction and Section 4.3).

**OTHER LINE BY LINE MINOR COMMENTS:**

Line 14: Replace "landforms" with "landscapes." Done

Line 18: briefly specify the origin of these badlands. Done

Line 82: Replace "paroxysmal." Also, specify some additional data about the mean annual rainfall regime (e.g., which months receive the highest rainfall?). Done (sections 2.1 and Figure 2).

Lines 123-124: this sentence is unclear. It seems that export values were calculated based on volumetric estimates from LiDAR campaigns, but the previous sentence suggests they were derived from hydro-sedimentary records. Please clarify. Done

Line 161: modes? Do you mean processes? Yes, corrected

Line 164: scour? Do you mean erosion? Yes, corrected

Line 176: which mass balance are you referring to? The one measured at the outlet? Rephrased

Lines 179-180: To better support this type of movement, could you provide an illustrative multitemporal profile? We're usure what the reviewer meant by this, sorry. However as previously said we reworked a little the introduction of the erosion hotspots labelling to clarify that they are mapped on a GIS software based on Figure 5 and 6, the DEM of the campaigns, the slopes and orthophotographs (2015 and 2021) that show might in particular show debris in areas labelles in blue as in this example :

orthophoto, then a zoom on the north-eastern part of Figure 5 where this crest failure took place and then what it looks like on the field. We clearly see debris on the second and last pictures that appear in Figure 5 as some positive signal.

To avoid overemphasizing this point, which is not the core of the presented work, and since we are unsure what the reviewer had in mind, we did not integrate these pictures in the paper.

Line 183: modes? Again, do you mean processes? Yes, corrected

Line 184: please, specify the time period Done

Lines 190-191: This concept requires clearer explanation. Done by reworking the whole paragraph

Line 197: did you observe a positive signal caused by this deposition? Sentence reworked to reference on Figure 5.

Line 200: do you mean a negative strong signal? It is not clear. Yes, clarified

Line 229: please, clarify what you mean with "emerging rock" changed to large boulders in the channel. They may show low specific drainage are as the are above water level even for intense rainfalls

Line 269; add a summary of rainfal pattern recorder over the 6 years time interval done, Figure 2.

Line 376: the 23% discrepancy should be briefly but more clearly explained in the Conclusion section. done

Appendix B: Units of measurement should be included throughout the appendix (in both the text and the table). We changed from density points to grams per cubic centimetre (g/cm³) throughout the paper.

Line 424: deposition beach unclear, please rephrase. Changed to sediment trap to correspond to previously used terminology

Table B1: is the "slope apron at the botton of the slope" a colluvial deposit? And the following category is the landslide body? Are the alluvial stocks referring to bars? This classification needs to be clarified from a geomorphological perspective. Corrected

Line 456: was the dry density perhaps maximal (rather than minimal) in late summer? Yes, corrected

| Before submission                                                             | After revision                                            |
|-------------------------------------------------------------------------------|-----------------------------------------------------------|
| Abstract                                                                      | Abstract                                                  |
| 1. Introduction                                                               | 1. Introduction                                           |
| 2. Site and Data                                                              | 2. Site and Data                                          |
| 2.1. The Draix-Laval experimental basin                                       | 2.1 The Draix-Laval experimental basin                    |
| 2.1.1. Draix-Laval's Terres Noires                                            | 2.1.1. The Laval's Terres Noires                          |
| 2.1.2. Draix-Bléone Critical Zone Observatory                                 | 2.1.2. Draix-Bléone Critical Zone Observatory             |
| 2.2. LiDAR Campaigns                                                          | 2.2. LiDAR Campaigns                                      |
| 2.2.1. UAV LiDAR survey (7 April 2015)                                        | 2.2.1. UAV LiDAR survey (7 April 2015)                    |
| 2.2.2. Airborne IGN LiDAR HD survey (23 June 2021)                            | 2.2.2. Airborne IGN LiDAR HD survey (23 June 2021)        |
|                                                                               | 2.3 Density measurements                                  |
| 3. Method                                                                     | 3. Method                                                 |
| 3.1. Outlet cumulative sediment export                                        | 3.1. LiDAR topographic change assessment                  |
| 3.2. Refinement of the co-registration of campaigns                           | 3.1.1. Refinement of the co-registration of the campaigns |
| 3.3. Diachronic analysis of local volume change                               | 3.1.2. Local distances between the point clouds           |
| 3.4. Effective marl density modelling                                         | 3.1.3. Inferring local volume changes                     |
|                                                                               | 3.2. Effective density model of the marls                 |
|                                                                               | 3.3. Outlet cumulative sediment export                    |
|                                                                               | 3.4. Hydrographic network reconstitution                  |
| 4. Results                                                                    | 4. Results                                                |
| 4.1. Mapping of erosion and deposition signals                                | 4.1. Erosion and deposition hotspots                      |
| 4.2. Contribution of erosion modes to sediment production                     | 4.2. Contributions to the sediment export                 |
| 4.3. Localisation of mass variations within the hydrographic network          | 4.3. Localisation within the hydrographic network         |
| 4.4. Production rate calculation for each specific drainage area compartments |                                                           |
| 5. Discussion                                                                 | 5. Discussion                                             |
| 5.1. Methodological constraints                                               | 5.1. Active mass wasting areas                            |
| 5.2. Spatial and temporal scales                                              | 5.2. Sediment production of the main geomorphological     |
| 5.3. Active mass wasting areas                                                | 5.3. Insights across spatial and temporal scales          |
| 5.4. Opportunities in a changing climate                                      | 5.4. Methodological constraints                           |
|                                                                               | 5.5. Opportunities in a changing climate                  |
| Conclusion                                                                    | Conclusion                                                |
| Appendices                                                                    | Appendices                                                |
| References                                                                    | References                                                |

**New references:**

- Wischmeier et al, 1998 (10.1029/TR039i002p00285)
- Leguédois et al, 2005 (10.2136/sssaj2005.0030)
- Legout et al, 2005 (10.1016/j.geoderma.2004.05.006)
- Stock and Dietrich, 2006 (10.1130/b25902.1)
- Le Bouteiller, 2025 (10.57745/DAEB1Z)
- Saez and al, 2011 (10.1002/esp.2141)
- Neugirg and al, 2016 (10.1016/j.geomorph.2016.06.027)
- Rovera and al, 2006 (10.7202/013735ar)
- Llena and al, 2020b (10.1002/esp.4822)
- Llena and al, 2020a (10.1002/esp.4821)
- Marsico and al, 2021 (10.3390/land10080828)
- Descroix and Gautier (10.1016/S0341-8162(02)00068-1)
- Moreno-de Las Heras, 2018 (10.1016/B978-0-12-813054-4.00002-2)
- Vergari and al, 2019 (10.1002/esp.4496)
- Montgomery and Foufoula-Georgiou, 1993 (doi.org/10.1029/93WR02463)
- Ballais, 1997 "Apparition et évolution de roubines à Draix », Cemagref
- Taille, 1996 « Taille des épandages torrentiels de la bordure méridionale du Dévoluy: rôle des héritages géomorphologiques »,
- De Ploey, 1991 "Bassins versants ravinés: analyses et prévisions selon le modèle Es", Bulletin de la Société géographique de Liège
- Vallauri, 1997 "Aperçu sur l'évolution écologique des forêts dans les préalpes du sud depuis la revolution"